 # FSI2P: A Hierarchical Focus–Sweep Registration Network with Dynamically Allocated Depth

**Zhixin Cheng** [1 2 3]  **Yujia Chen** [1 2]  **Xujing Tao** [1 2]  **Bohao Liao** [1 2]  **Xiaotian Yin** [4]  **Baoqun Yin** [1]  **Tianzhu Zhang** [1 2]

## Abstract

Image-to-point cloud registration is often challenged by viewpoint changes, cross-modal discrepancies, and repetitive textures, which induce scale ambiguity and consequently lead to erroneous correspondences. Recent detection-free methods alleviate this issue by leveraging multi-scale features and transformer-based interactions. However, they still suffer from attention drift across layers and intra-scale inconsistencies, hindering precise registration. Inspired by complex scene observation, we propose a "Focus–Sweep" paradigm and develop a Hierarchical Mamba Interaction Module within an SSM-based framework to enhance multi-level cross-modal feature association. In addition, we introduce a Dynamic Layer Allocation Strategy that adaptively determines the iteration depth to better exploit geometric constraints and improve matching robustness. Extensive experiments and ablations on two benchmarks, RGB-D Scenes V2 and 7-Scenes, demonstrate that our approach achieves state-of-the-art performance.

## 1. Introduction

Image-to-point cloud registration (I2P) task aims to estimate the rigid transformation that aligns a 3D point cloud with a 2D image of the same scene. It typically involves establishing cross-modal correspondences between image pixels and 3D points, followed by pose estimation to recover rotation

[1] School of Information Science and Technology, University of Science and Technology of China [2] National Key Laboratory of Deep Space Exploration, Deep Space Exploration Laboratory, University of Science and Technology of China [3] School of Computer Science and Information Engineering, Hefei University of Technology [4] Institute of Advanced Technology, University of Science and Technology of China. Correspondence to: Tianzhu Zhang <tzzhang@ustc.edu.cn>.

*Proceedings of the 43rd International Conference on Machine Learning*, Seoul, South Korea. PMLR 306, 2026. Copyright 2026 by the author(s).

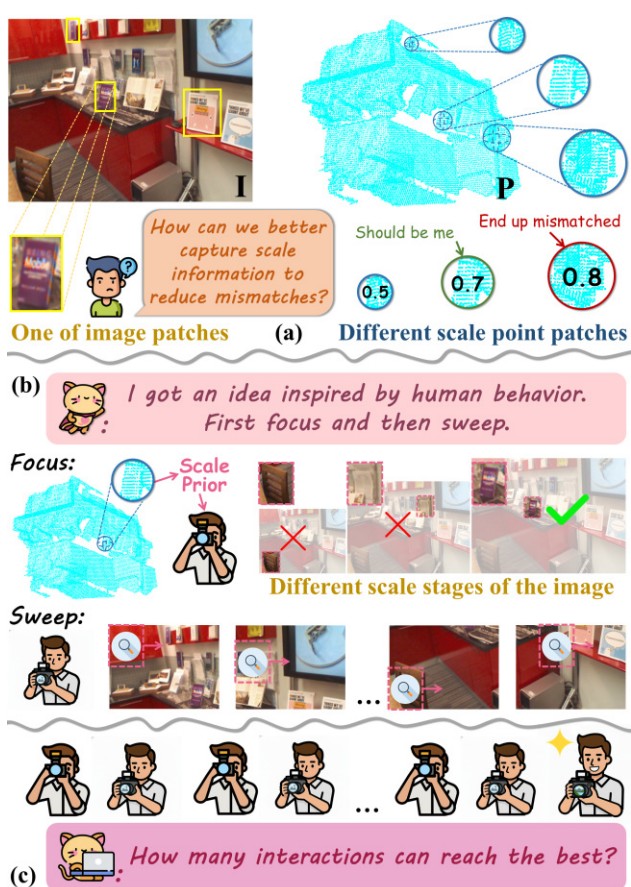

*Figure 1.* (a) Visualization of incorrect matches caused by scale discrepancies between the image and point cloud; black numbers indicate similarity scores. (b) Visualization of our *Focus–Sweep* pipeline. *Focus* aligns scale ranges across different levels, while *Sweep* performs block-wise fine-grained interactions. (c) Visualization of the challenge in choosing the number of interaction layers.

and translation. This task plays a critical role in a wide range of applications, including 3D reconstruction(Mouragnon et al., 2006a; Geiger et al., 2011; Mouragnon et al., 2006b; Deng et al., 2024; 2025; Liao et al., 2024), SLAM(Durrant-Whyte & Bailey, 2006; Khairuddin et al., 2015; Deng et al., 2023; Zuo et al., 2017; Chen et al., 2025; Li et al., 2026), and visual localization(Bolognini et al., 2005; Schönberger et al.,

2018; He et al., 2023; Liu et al., 2026). However, variations in sensor perspectives and the fundamental differences between dense, grid-structured images and sparse, unordered point clouds lead to scale ambiguity, making cross-modal association and fusion challenging and still unresolved.

To address the image-to-point cloud registration task, existing methods can be broadly categorized into two groups. The first, detect-then-match(Feng et al., 2019; Wang et al., 2021; Ren et al., 2022) paradigm, separately detects 2D and 3D keypoints and matches them using semantic features. However, the modality gap between texture-based images and geometry-based point clouds makes it difficult to obtain repeatable keypoints and consistent descriptors. To overcome these limitations, recent detection-free approaches, such as 2D3D-MATR(Li et al., 2023), employ a coarse-to-fine framework that begins with patch-level matching, refines pixel-level correspondences, and finishes with pose estimation via PnP-RANSAC(Lepetit et al., 2009; Fischler & Bolles, 1981). Nevertheless, in challenging scenarios like repetitive patterns and large viewpoint variations, scale ambiguity may still lead to inaccurate cross-modal matching.

Building upon the discussion above, we identify a key challenge that remains crucial for achieving accurate and robust image-to-point cloud matching: ***How to reduce mismatches caused by scale ambiguity?*** As shown in Figure 1(a), errors in matching may arise from differences in sensor viewpoints and from intrinsic disparities between images and point clouds. To tackle this issue, B2-3D(Cheng et al., 2025b) has made progress by using hierarchical cross-attention to interact with point clouds. However, through our experimental validation, we identified two issues that require further attention. First, the cross-modal registration task inherently requires transformer-based interactions, and deep stacking of cross-attention layers may amplify early attention biases, increasing alignment noise due to the Matthew effect. Second, although multi-scale designs help alleviate scale differences, they still struggle with scale ambiguity caused by repetitive textures at different resolutions. These issues are likely to remain unresolved without the exploration of new interaction paradigms.

To tackle the challenges mentioned above, we draw inspiration from the progressive perception process in complex scenes, as illustrated in Figure 1(b). Specifically, global contextual cues are first estimated to provide coarse structural guidance, followed by local refinement to identify fine-grained correspondences across modalities. Inspired by this process, we propose a cross-modal aggregation model, **Focus-Sweep**, which alternates between global calibration and local interaction to progressively enhance image–point cloud feature correlations.

AS SaFiRe(Mao et al., 2025) proposes a novel paradigm for employing Mamba, as it leverages state-space sequence

modeling (SSM) to emulate a coarse-to-fine cognitive process, alternating between global semantic alignment and local region-wise refinement to progressively enhance cross-modal understanding. Building upon this inspiration, we use Mamba (Gu & Dao, 2024) as the underlying architecture to support our framework in a cognitively inspired, structurally efficient manner. Specifically, the focus operation quickly scans both the point cloud and the image, establishing rough correspondences between modalities by modulating multi-level image features with the point cloud's general scale after the SSM scan. The sweep operation refines these coarse results by carefully inspecting each image region and repeatedly rechecking the point cloud. This operation alternates the SSM scan between each image region and the point cloud, allowing Mamba to concentrate locally while leveraging its sequential sensitivity and memory-efficient design to maintain global context. Moreover, through the repetition of this two-stage process, the model incrementally improves its cross-modal understanding and accurately identifies correspondences.

This naturally leads to the second issue (as shown in Figure 1(c)): ***How many iterations are necessary to achieve the optimal result?*** Given that selecting the number of interaction layers is a discrete, nondifferentiable decision, it cannot be directly learned by conventional gradient-based optimization. We propose leveraging reinforcement learning(Kaelbling et al., 1996), using global registration constraints as rewards, to dynamically and adaptively select the number of layers. This mirrors complex scene observation, where observation iterations are not predetermined but instead continue until a certain goal is achieved (such as finding the corresponding match or confirming its absence).

In summary, our work can be summarized as follows:

- We propose a novel Hierarchical Focus–Sweep Registration Network with Dynamically Allocated Depth (***FS-I2P***), which demonstrates excellent accuracy and strong generalization capabilities in the image-to-point cloud registration task.
- We use the Hierarchical Mamba Interaction Module to address multi-level correlation modeling between image and point cloud features with a Mamba-based interaction paradigm. Besides, we introduced the Dynamic Layer Allocation Strategy, optimizing the iteration depth by dynamically selecting the number of iterations, thereby improving the robustness of cross-modal matching.
- Extensive experiments and ablation studies on two benchmarks, RGB-D Scenes V2 and 7-Scenes, demonstrate the superiority of the proposed network, establishing it as the state-of-the-art method for image-to-point cloud registration tasks.

**Conflict of Interest Disclosure.**   The authors declare no financial conflicts of interest related to this work.

## 2. Related Works

In this section, we provide a brief overview of related works in image-to-point cloud (I2P) registration, discussing stereo image registration, point cloud registration, and inter-modality registration techniques.

### 2.1. Image and Point Cloud Registration

Traditional stereo image registration methods often rely on detector-based techniques, such as SIFT (Ng & Henikoff, 2003) and ORB (Rublee et al., 2011), to match features between images. With the advent of deep learning, methods like SuperGlue (Sarlin et al., 2020) have enhanced matching accuracy by employing transformers (Vaswani et al., 2017). However, these methods face challenges in texture-less regions, where low keypoint repeatability undermines robustness. To mitigate this, detector-free methods, such as LoFTR (Sun et al., 2021) and Efficient LoFTR (Wang et al., 2024b), utilize coarse-to-fine strategies and global receptive fields to establish dense and reliable correspondences. Point cloud registration has evolved from traditional handcrafted descriptors like PPF (Moheimani et al., 2006) and FPFH (Rusu et al., 2009) to deep learning-based approaches. CoFiNet (Yu et al., 2021) introduced a coarse-to-fine pipeline for detector-free registration. Recent methods, such as GeoTransformer (Qin et al., 2023), replace traditional estimators like RANSAC (Fischler & Bolles, 1981) with learning-based techniques, offering faster and more accurate results. GeoTransformer also employs transformers to model global dependencies and introduces a local-to-global framework for precise alignment, eliminating the need for RANSAC.

### 2.2. Inter-Modality Registration

Given a 3D point $\mathbf{X} \in \mathbb{R}^3$, its projection onto the image plane is defined as:

$$\mathbf{x} \sim \mathbf{K}(\mathbf{R}\mathbf{X} + \mathbf{t}) \tag{1}$$

where $\mathbf{K}$ denotes the camera intrinsic matrix, $\mathbf{R} \in \mathrm{SO}(3)$ and $\mathbf{t} \in \mathbb{R}^3$ represent the camera rotation and translation, respectively, and $\mathbf{x} \in \mathbb{R}^2$ is the corresponding pixel coordinate in the image. Based on this projection model, the image-to-point registration task aims to establish correspondences between image pixels and 3D points, $\mathbf{x} \leftrightarrow \mathbf{X}$, which serves as a fundamental step for cross-modal matching and registration.

Inter-modality registration is more challenging than intra-modality registration due to the significant domain differences between image and point cloud data. Traditional

methods typically follow a detect-then-match paradigm, such as 2D3DMatch-Net (Feng et al., 2019), which uses SIFT (Ng & Henikoff, 2003) for keypoint extraction, or ISS (Sontag, 1998), which creates local patches on point clouds and utilizes CNNs and PointNet (Qi et al., 2017) for matching. P2-Net (Wang et al., 2021) improves efficiency by simultaneously detecting and matching keypoints. Camera relocalization has also been investigated in this context. DSAC (Brachmann et al., 2017) formulates a differentiable RANSAC pipeline for end-to-end camera localization, whereas Li et al. (Li et al., 2020) develop a hierarchical classification-and-regression scheme to enhance the accuracy and robustness of scene coordinate estimation. However, keypoint detection accuracy often suffers in cross-modal settings (Simon et al., 2017; Barroso-Laguna et al., 2019), leading to the development of detector-free methods. For instance, 2D3D-MATR (Li et al., 2023) adopts a coarse-to-fine approach with a transformer network and PnP-RANSAC to refine results, eliminating the need for keypoint detection. B2-3D (Cheng et al., 2025b) enhances registration through uncertainty modeling and domain adaptation. CA-I2P (Cheng et al., 2025c) aligns image and point cloud features from a channel perspective, while Diff$^2$I2P (Mu et al., 2025) bridges the modality gap with a depth-conditioned diffusion model. Flow-I2P (An et al., 2025) improves manifold alignment using a Beltrami flow-based approach. Our method, *FS-I2P*, innovatively draws inspiration from complex scene observation behavior, leveraging the focus-sweep pattern to aggregate image and point cloud modalities. It also utilizes reinforcement learning to adaptively adjust the number of layers, significantly boosting image-to-point cloud registration performance.

## 3. Method

Let $I \in \mathbb{R}^{H \times W \times 3}$ and $P \in \mathbb{R}^{N \times 3}$ represent an image and a point cloud from the same scene, respectively, where $H$ and $W$ denote the height and width of the image, and $N$ is the number of 3D points in the point cloud. The goal of image-to-point cloud registration is to estimate a rigid transformation $[R, \mathbf{t}]$, where $R \in \mathrm{SO}(3)$ is the rotation matrix, and $\mathbf{t} \in \mathbb{R}^3$ is the translation vector (Cheng et al., 2026c; 2025a).

Our method *FS-I2P* (see in Figure 2), innovatively enables a multi-level Mamba interaction inspired by complex scene observation behavior and resolves the non-differentiable layer selection issue through reinforcement learning, significantly improving the accuracy and robustness of image-to-point cloud registration. Firstly, in the Hierarchical Mamba Interaction Module, we use Mamba as the underlying architecture to support our cognitively inspired framework, where the focus operation establishes rough correspondences by scanning both the point cloud and image, and the sweep

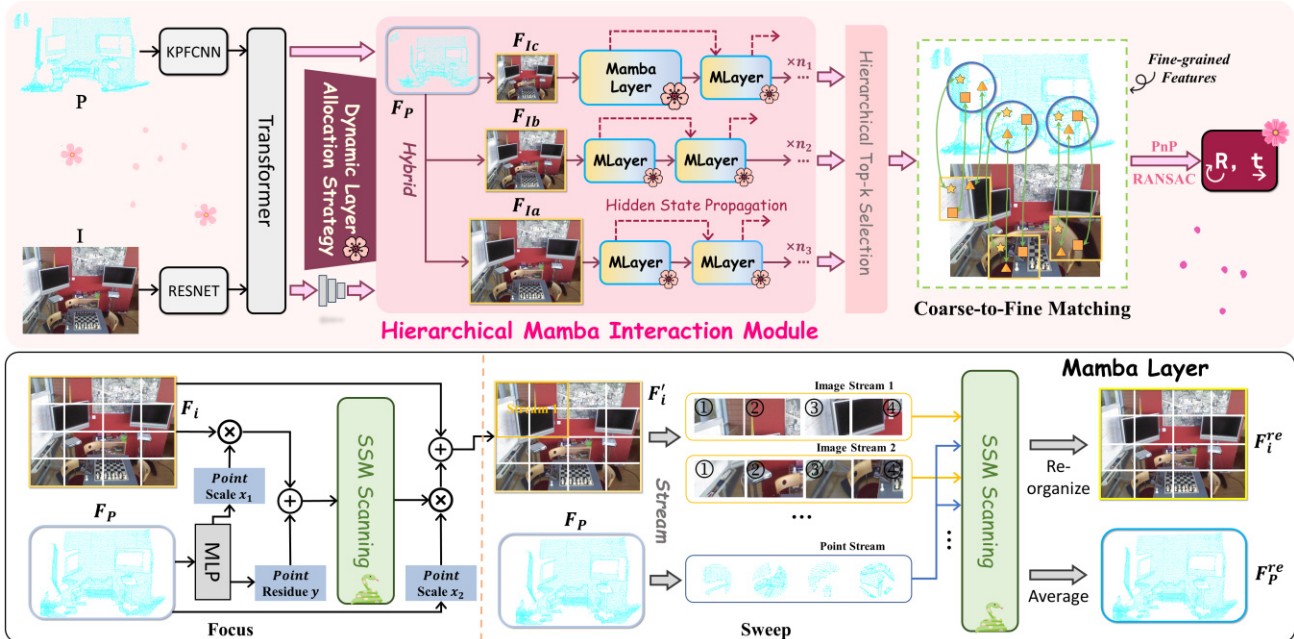

*Figure 2.* Overall pipeline of FS-I2P. We extract image and point cloud features, and then perform a initial interaction step implemented with transformer layers. In our Hierarchical Mamba Interaction Module, built on Mamba paradigm (Mao et al., 2025), Focus performs a global scan and updates image features under point-cloud scales, while Sweep iteratively refines them via local scans for more accurate cross-modal matching. Moreover, our Dynamic Layer Allocation Strategy uses reinforcement learning with global constraints as rewards to adaptively choose the number of interaction layers. The recovered multi-level point cloud and image features generate a score map via cosine similarity and max operations, enabling coarse-level matching and refining fine-level correspondences. Finally, PnP+RANSAC is applied to estimate the rigid transformation.

operation refines these results through iterative local scans, enabling the model to improve cross-modal understanding and accurately identify correspondences. Moreover, the Dynamic Layer Allocation Strategy employs reinforcement learning with global constraints as rewards to adaptively determine the number of interaction layers, resembling complex scene observation behavior where observations persist until a goal is achieved, rather than being limited to a fixed number of iterations. In the subsequent coarse-to-fine matching process, a coarse matching set $M_c$ is first obtained through cosine similarity calculation (Strehl et al., 2000). High-resolution image and point cloud features are then employed to refine the dense matching set $M_f$ from patch matching pairs. Finally, the rigid transformation is accurately estimated using the PnP+RANSAC algorithm.

### 3.1. Hierarchical Mamba Interaction Module

We adopt ResNet (He et al., 2016) with an FPN (Lin et al., 2017) and KPFCNN (Thomas et al., 2019) to extract 2D and 3D features with positional encoding, respectively. At the lowest resolution, the image features are denoted as $\mathbf{F}_I \in \mathbb{R}^{H \times W \times C}$, and the point cloud features as $\mathbf{F}_P \in \mathbb{R}^{N \times C}$. To bridge the domain gap between images and point clouds, these features are further refined using self- and cross-attention transformers, facilitating cross-modal

feature alignment. However, due to repetitive textures and differences in detector viewpoints within the scene, scale ambiguity can arise, thereby affecting matching accuracy.

**Mamba Layer.** To address these challenges, we studied how complex scene observation and identify correspondences between different modalities in complex scenes. Inspired by SaFiRe (Mao et al., 2025), we use a two-stage interaction approach: First, *Focus*, which involves an initial assessment of the point cloud scale and establishing correlations across multi-level images; second, *Sweep*, which focuses on finer details by re-examining the point cloud and inspecting image regions to extract relevant information. By alternating these two operations across multiple layers, the model gradually enhances its cross-modal understanding. This cross-layer focus-sweep repetition serves as the core mechanism for improving matching accuracy through precise interaction. To better understand this process, we provide a detailed description of the design of the focus and sweep operations in two parts. The hidden state is propagated through the same operation to provide prior information, while the number of Mlayers is controlled by the Dynamic Layer Allocation Strategy (shown as the pink flower with a black outline in the Figure 2). Image features first pass through three layers of a lightweight CNN to form hierarchical features $F_{Ia}, F_{Ib}, F_{Ic}$, which, along with point cloud

features $F_P$, serve as the input to the Focus-Sweep Layers.

**I. *Focus:*** The Focus operation introduces a global scale prior to guide the early interaction between image and point cloud features. The point cloud feature $F_P$ is first summarized by an MLP to generate point-conditioned modulation factors, including the first point scale $x_1$, the point residue $y$, and the second point scale $x_2$. These factors are used to globally calibrate the image feature $F_i$ before and after SSM scanning.

Specifically, the first point scale $x_1$ modulates the image feature at the input side, allowing the image representation to be adjusted according to the global structural scale of the point cloud. The point residue $y$ is then added to the modulated image feature, injecting point-aware geometric context into the image branch. After this point-guided adaptation, the feature is processed by the SSM scanning layer, where long-range spatial dependencies are further modeled. Finally, the second point scale $x_2$ is applied to the scanned feature, and the result is combined with the original image feature through a residual connection and get $F_i'$.

Through this design, Focus performs lightweight global calibration rather than dense cross-modal matching. The point-derived scale and residue factors provide coarse geometric guidance for the image feature, producing a scale-aware representation that is better prepared for the subsequent Sweep operation, where fine-grained image–point cloud correspondences are progressively refined.

**II. *Sweep:*** The Sweep operation performs fine-grained cross-modal interaction by progressively propagating local structural cues between image and point cloud features, thereby refining correspondence estimation beyond the global scale prior introduced by Focus.

Therefore, we build the Sweep operation upon the recent VMamba architecture and design a stream-wise cross-modal interaction scheme for fine-grained image-to-point cloud matching. Different from global token mixing, Sweep reorganizes the image feature into multiple local image streams and repeatedly couples them with the point cloud stream. In this way, each local image region can interact with the shared 3D geometric context during the VSSM scanning process, enabling the hidden states of SSM to progressively absorb region-specific visual cues and point-wise structural priors.

Given the image feature $F_I \in \mathbb{R}^{h \times w \times C}$ and the point feature $F_P \in \mathbb{R}^{N \times C}$, we first reshape $F_I$ into $t$ local image streams, denoted as $\mathcal{F}_I = \{F_I^1, F_I^2, \cdots, F_I^t\}$. Each local stream $F_I^u$ contains $o^2$ image tokens with $C$-dimensional features, i.e., $F_I^u \in \mathbb{R}^{o^2 \times C}$, where $o$ denotes the local stream size and $t = hw/o^2$ is the number of image streams.

Instead of processing the image and point cloud separately,

we construct a hybrid multi-modal stream by inserting the point stream after each local image stream. Specifically, the hybrid feature stream $F_H$ is formed by alternately concatenating each local image stream $F_I^u$ with the point feature $F_P$, resulting in the order of $F_I^1, F_P, F_I^2, F_P, \cdots, F_I^t, F_P$. The resulting stream has a feature dimension of $(hw + tN) \times C$, where $t$ is the number of local image streams and $N$ is the number of point tokens. The hybrid stream $F_H$ is then fed into the VSSM layer to obtain the updated hybrid representation $F_H'$.

Through this sequential modeling process, the point cloud tokens are repeatedly introduced as geometric anchors for different image regions. This design allows the SSM hidden states to adapt to the image–point cloud registration paradigm, where local visual patterns need to be progressively associated with global 3D structural cues. As a result, Sweep can conduct fine-grained cross-modal interaction while preserving the long-range modeling ability of SSM.

After VSSM scanning, the processed hybrid stream $F_H'$ is separated according to the original alternating image–point order. Specifically, the segments corresponding to local image streams are extracted as the updated image streams $F_I^{1'}, F_I^{2'}, \cdots, F_I^{t'}$, while the repeated point-stream segments are extracted as multiple updated point-stream instances $F_P^{1'}, F_P^{2'}, \cdots, F_P^{t'}$.

The image streams are reorganized back to the spatial layout with the size of $h \times w \times C$:

$$F_I^{re} = \text{Reorganize}\left(F_I^{1'}, F_I^{2'}, \cdots, F_I^{t'}\right). \quad (2)$$

For the point cloud feature, since the point stream interacts with different image regions during the scanning process, we aggregate all updated point-stream instances to obtain the refined point representation $F_P^{re}$. Specifically, $F_P^{re}$ is computed as the weighted summation of all updated point-stream instances $F_P^{u'}$, where $\lambda_u$ denotes the learnable weight for the $u$-th point-stream instance. The refined point feature $F_P^{re}$ has the size of $N \times C$. The resulting $F_I^{re}$ and $F_P^{re}$ are then forwarded to the next MLayer for subsequent refinement.

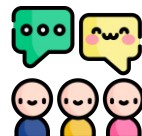

***Discussion.***

***Why not use transformer for direct interaction at different scales?*** Although transformers are commonly used for feature aggregation, Mamba not only retains the advantages discussed above but also mitigates an overlooked issue. As shown in Figure 3, increasing attention layers first improves image–point alignment (lower MMD), but further stacking causes MMD to rise. We attribute this to a Matthew-effect-like bias in deep cross-attention: small early-layer attention

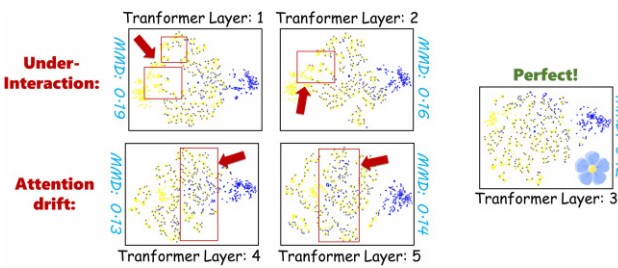

*Figure 3.* T-SNE plots and the corresponding MMD values under different transformer depths suggest that too few layers yield insufficient cross-modal interaction, while too many layers cause attention drift. This motivates using SSM as a potentially better and more stable alternative for multi-scale interaction.

deviations can be amplified by repeated global re-alignment in the absence of explicit geometric constraints, causing the model to over-focus on a few regions and introducing alignment noise that increases the distribution gap between image and point features, ultimately weakening geometric consistency. Nevertheless, multi-level top-$k$ selection remains beneficial by enlarging the candidate set across scales under this condition and increasing the chance of correct correspondences.

### 3.1.1. HIERARCHICAL TOP-K SELECTION

After the MLayer interaction, we obtain refined multi-scale image features $\{F_{I_a}^{re}, F_{I_b}^{re}, F_{I_c}^{re}\}$ and point cloud features $\{F_{P_a}^{re}, F_{P_b}^{re}, F_{P_c}^{re}\}$.

We first concatenate the image features into a unified stream:

$$F_I^{re} = \text{concat}(F_{I_a}^{re}, F_{I_b}^{re}, F_{I_c}^{re}). \qquad (3)$$

Then, cosine similarity is computed between the unified image features and each scale of point cloud features:

$$S_k = \cos(F_I^{re}, F_{P_k}^{re}), \quad k \in \{a, b, c\}. \qquad (4)$$

The three initial score maps are aggregated by element-wise maximization to obtain the final score map:

$$S = \max(S_a, S_b, S_c). \qquad (5)$$

Finally, patch-level matching pairs are obtained by selecting the top-$k$ entries from the score map $S$.

### 3.2. Dynamic Layer Allocation Strategy

The number of Mamba interaction layers is crucial for image-to-point cloud registration: insufficient interactions may lead to ambiguous correspondences, while excessive interactions introduce redundant computation and may over-refine noisy matches. Inspired by complex scene observation behavior, observation iterations are not predetermined

but continue until the goal is reached, either locating the corresponding match or confirming that no match exists. Thus instead of adopting a fixed interaction depth, we propose a reinforcement learning (RL)-based dynamic layer allocation strategy that adaptively determines the interaction depth conditioned on the current matching status.

Specifically, for the three hierarchical scales, we denote the selected numbers of MLayer iterations as $\{n_1, n_2, n_3\}$, and $l_{\max}$ is the maximum allowed depth. Notably, we also allow skipping interaction at a certain level, since when the overall scene structure is dominated by a particular scale, performing interactions at other scales may introduce unnecessary interference. For simplicity, we use $n$ to represent the interaction depth at an arbitrary scale in the following.

Given image and point tokens, we construct a compact state vector $s$ by concatenating their mean and max-pooled statistics.

A lightweight policy network predicts a categorical distribution over the candidate interaction depths:

$$\mathbf{z} = g_\theta(s) \in \mathbb{R}^A, \quad \pi_\theta(n \mid s) = \text{Softmax}(\mathbf{z}), \qquad (6)$$

where each action determines the interaction depth $n$. During training, an action is sampled from the policy:

$$a \sim \pi_\theta(\cdot \mid s) \qquad (7)$$

while greedy selection is used at inference time:

$$a = \arg\max \mathbf{z}, \qquad (8)$$

And for each decision, we record the log-probability:

$$\log p = \log \pi_\theta(a \mid s). \qquad (9)$$

We use the matching constraint $L_i$ as supervision and define the reward as:

$$R = \frac{1}{L_i + \delta}, \qquad (10)$$

where $\delta$ is a small constant. To optimize the discrete policy, we apply REINFORCE:

$$\nabla_\theta J(\theta) = \mathbb{E}\big[\nabla_\theta \log p \cdot (R - B)\big], \qquad (11)$$

where $B$ is an exponential moving-average baseline of the reward to reduce variance, updated by

$$B \leftarrow (1 - \epsilon) B + \epsilon R, \qquad (12)$$

with momentum coefficient $\epsilon \in (0, 1)$. Accordingly, the reinforcement loss is:

$$L_r = -(R - B) \log p. \qquad (13)$$

This strategy enables the model to adaptively allocate the interaction depth $n$ at each scale conditioned on the current cross-modal matching status, improving robustness while reducing sensitivity to fixed hyperparameters.

*Table 1.* Evaluation results on RGB-D Scenes V2 and 7-Scenes. **Orange** and **Purple** numbers highlight the best, the second best are **Boldfaced** and the baseline are underlined.

| Dataset | RGB-D Scenes V2 | | | | | 7-Scenes | | | | | | | |
|---|---|---|---|---|---|---|---|---|---|---|---|---|---|
| Model | Scene-11 | Scene-12 | Scene-13 | Scene-14 | Mean | Chess | Fire | Heads | Office | Pumpkin | Kitchen | Stairs | Mean |
| Mdpt(m) | 1.74 | 1.66 | 1.18 | 1.39 | 1.49 | 1.78 | 1.55 | 0.80 | 2.03 | 2.25 | 2.13 | 1.84 | 1.49 |
| *Inlier Ratio ↑* | | | | | | | | | | | | | |
| FCGF-2D3D(Choy et al., 2019) | 6.8 | 8.5 | 11.8 | 5.4 | 8.1 | 34.2 | 32.8 | 14.8 | 26 | 23.3 | 22.5 | 6.0 | 22.8 |
| P2-Net(Wang et al., 2021) | 9.7 | 12.8 | 17.0 | 9.3 | 12.2 | 55.2 | 46.7 | 13.0 | 36.2 | 32.0 | 32.8 | 5.8 | 31.7 |
| Predator-2D3D(Huang et al., 2021b) | 17.7 | 19.4 | 17.2 | 8.4 | 15.7 | 34.7 | 33.8 | 16.6 | 25.9 | 23.1 | 22.2 | 7.5 | 23.4 |
| 2D3D-MATR(Li et al., 2023) | 32.8 | 34.4 | 39.2 | 23.3 | 32.4 | 72.1 | 66.0 | 31.3 | 60.7 | 50.2 | 52.5 | 18.1 | 50.1 |
| B2-3Dnet(Cheng et al., 2025b) | 36.4 | 32.7 | 43.8 | 27.4 | 35.1 | 73.8 | 66.7 | 33.1 | 61.7 | 50.8 | 52.3 | 18.1 | 50.9 |
| CA-I2P(Cheng et al., 2025c) | 38.6 | 40.6 | 38.9 | 24.0 | 35.5 | 73.6 | 66.4 | 34.5 | 62.4 | 52.1 | 52.8 | 19.1 | 51.6 |
| Diff²I2P(Mu et al., 2025) | - | - | - | - | 36.9 | 74.1 | 68.8 | 39.2 | 65.6 | 52.1 | 54.2 | 18.1 | 53.2 |
| Flow-I2P(An et al., 2025) | 49.6 | 44.0 | 36.5 | 30.4 | 40.1 | 76.6 | 64.7 | 37.1 | 62.0 | 52.3 | 52.8 | 18.5 | 52.0 |
| FS-I2P(Ours) | 50.4 | 49.7 | 40.4 | 31.2 | 42.9 | 75.7 | 68.6 | 39.3 | 66.3 | 52.9 | 55.3 | 19.4 | 53.9 |
| *Feature Matching Recall ↑* | | | | | | | | | | | | | |
| FCGF-2D3D(Choy et al., 2019) | 11.1 | 30.4 | 51.5 | 15.5 | 27.1 | 99.7 | 98.2 | 69.9 | 97.1 | 83.0 | 87.7 | 16.2 | 78.8 |
| P2-Net(Wang et al., 2021) | 48.6 | 65.7 | 82.5 | 41.6 | 59.6 | 100.0 | 99.3 | 58.9 | 99.1 | 87.2 | 92.2 | 16.2 | 79 |
| Predator-2D3D(Huang et al., 2021b) | 86.1 | 89.2 | 63.9 | 24.3 | 65.9 | 91.3 | 95.1 | 76.6 | 88.6 | 79.2 | 80.6 | 31.1 | 77.5 |
| 2D3D-MATR(Li et al., 2023) | 98.6 | 98.0 | 88.7 | 77.9 | 90.8 | 100.0 | 99.6 | 98.6 | 100.0 | 92.4 | 95.9 | 58.2 | 92.1 |
| B2-3Dnet(Cheng et al., 2025b) | 100.0 | 99.0 | 92.8 | 85.8 | 94.4 | 100.0 | 100.0 | 98.6 | 100.0 | 92.7 | 95.6 | 64.9 | 93.1 |
| CA-I2P(Cheng et al., 2025c) | 100.0 | 100.0 | 91.8 | 82.7 | 93.6 | 100.0 | 100.0 | 98.6 | 100.0 | 92.0 | 95.5 | 60.8 | 92.4 |
| Diff²I2P(Mu et al., 2025) | - | - | - | - | 77.1 | 100.0 | 100.0 | 100.0 | 100.0 | 93.4 | 96.2 | 55.4 | 92.2 |
| Flow-I2P(An et al., 2025) | 100.0 | 100.0 | 94.5 | 78.7 | 93.3 | 100.0 | 99.7 | 95.1 | 99.9 | 93.1 | 96.8 | 56.7 | 91.6 |
| FS-I2P(Ours) | 100.0 | 100.0 | 92.8 | 85.0 | 94.4 | 100.0 | 100.0 | 98.6 | 100.0 | 92.0 | 96.3 | 60.2 | 92.4 |
| *Registration Recall ↑* | | | | | | | | | | | | | |
| FCGF-2D3D(Choy et al., 2019) | 26.5 | 41.2 | 37.1 | 16.8 | 30.4 | 89.5 | 79.7 | 19.2 | 85.9 | 69.4 | 79.0 | 6.8 | 61.4 |
| P2-Net(Wang et al., 2021) | 40.3 | 40.2 | 41.2 | 31.9 | 38.4 | 96.9 | 86.5 | 20.5 | 91.7 | 75.3 | 85.2 | 4.1 | 65.7 |
| Predator-2D3D(Huang et al., 2021b) | 44.4 | 41.2 | 21.6 | 13.7 | 30.2 | 69.6 | 60.7 | 17.8 | 62.9 | 56.2 | 62.6 | 9.5 | 48.5 |
| 2D3D-MATR(Li et al., 2023) | 63.9 | 53.9 | 58.8 | 49.1 | 56.4 | 96.9 | 90.7 | 52.1 | 95.5 | 80.9 | 86.1 | 28.4 | 75.8 |
| B2-3Dnet(Cheng et al., 2025b) | 58.3 | 60.8 | 74.2 | 60.2 | 63.4 | 98.3 | 90.5 | 56.2 | 96.4 | 84.0 | 86.1 | 32.4 | 77.7 |
| CA-I2P(Cheng et al., 2025c) | 68.1 | 73.5 | 63.9 | 47.8 | 63.3 | 99.0 | 90.7 | 68.5 | 96.2 | 83.0 | 88.1 | 31.1 | 79.5 |
| Diff²I2P(Mu et al., 2025) | - | - | - | - | 60.5 | 99.0 | 95.6 | 74.0 | 98.9 | 86.8 | 90.2 | 36.5 | 83.0 |
| Flow-I2P(An et al., 2025) | 90.0 | 65.9 | 54.8 | 63.0 | 68.4 | 98.8 | 90.0 | 58.4 | 93.9 | 82.1 | 88.6 | 37.6 | 78.4 |
| FS-I2P(Ours) | 71.1 | 81.4 | 70.9 | 68.1 | 72.9 | 98.3 | 94.9 | 86.3 | 98.0 | 80.3 | 91.2 | 29.7 | 82.7 |
| FS-I2P$_{+Dino\ v2}$ | 90.3 | 89.2 | 91.8 | 73.9 | 86.3 | 99.7 | 96.0 | 91.8 | 98.7 | 81.6 | 92.1 | 32.4 | 84.6 |

## 3.3. Model Training & Inference

To obtain the total loss, first let us examine the loss functions for the coarse and fine matching networks. Both $L_{coarse}$ and $L_{fine}$ use the general circle loss (Sun et al., 2020; Qin et al., 2022). Given an anchor descriptor $d_i$, the descriptors of its positive and negative pairs are $\mathcal{D}_i^P$ and $\mathcal{D}_i^N$, respectively:

$$L_i = \frac{1}{\zeta} \log \left[ 1 + \left( \sum_{d^j \in \mathcal{D}_i^P} e^{\beta_p^{i,j}(d_i^j - \Delta_P)} \right) \cdot \left( \sum_{d^k \in \mathcal{D}_i^N} e^{\beta_n^{i,k}(\Delta_n - d_i^k)} \right) \right], \quad (14)$$

where $d_i^j$ is the $L_2$ feature distance, and the individual weights for the positive and negative pairs are defined as

$$\beta_p^{i,j} = \zeta \lambda_p^{i,j}(d_i^j - \Delta_p), \quad (15)$$

$$\beta_n^{i,k} = \zeta \lambda_n^{i,k}(\Delta_n - d_i^k), \quad (16)$$

with $\lambda_p^{i,j}$ and $\lambda_n^{i,k}$ as scaling factors (Huang et al., 2021a). Combining the discussions above, the total loss is composed of two key components: the matching loss $L_i$ (including $L_{coarse}$ and $L_{fine}$) from the matching process. The total loss is computed as:

$$L_{total} = \xi_1 L_i + \xi_2 L_r, \quad (17)$$

where $\xi_i$ are hyperparameters balancing the contribution of different loss terms.

*Table 2.* Ablation studies of modules on RGB-D Scenes V2. HTS stands for hierarchical top-k selection, and DLAS stands for dynamic layer allocation strategy.

| Exp. | Focus | Sweep | HTS | DLAS | IR↑ | FMR↑ | RR↑ |
|---|---|---|---|---|---|---|---|
| M1 | ✗ | ✗ | ✓ | ✗ | 32.4 | 90.8 | 56.4 |
| M2 | ✗ | ✗ | ✗ | ✗ | 31.3 | 85.6 | 49.5 |
| M3 | ✗ | ✓ | ✓ | ✗ | 36.8 | 92.5 | 62.4 |
| M4 | ✓ | ✓ | ✓ | ✗ | 38.5 | 93.8 | 68.2 |
| M5 | ✓ | ✗ | ✓ | ✗ | 35.1 | 91.2 | 59.7 |
| M6 | ✓ | ✓ | ✗ | ✓ | 36.2 | 92.4 | 56.2 |
| M7 | ✓ | ✓ | ✓ | ✓ | **40.1** | **94.4** | **72.9** |

# 4. Experiments

## 4.1. Datasets and Implementation Details

Based on the 2D3D-MATR benchmark, we conducted extensive experiments and ablation studies on two challenging benchmarks: RGB-D Scenes V2 (Lai et al., 2014) and 7-Scenes (Glocker et al., 2013).

**Dataset.** *RGB-D Scenes V2* consists of 14 scenes containing furniture. For each scene, point cloud fragments are generated from every 25 consecutive depth frames, and one RGB image is sampled per 25 frames. We select image-point-cloud pairs with an overlap ratio of at least 30%. Scenes 1-8 are used for training, scenes 9-10 for validation, and scenes 11-14 for testing, resulting in 1,748 training pairs, 236 validation pairs, and 497 testing pairs.

*7-Scenes* is a collection of tracked RGB-D camera frames. All seven indoor scenes were recorded using a handheld Kinect RGB-D camera at a resolution of 640×480. Image-to-point-cloud pairs with at least 50% overlap are selected from each scene, adhering to the official stream split for training, validation, and testing. This results in 4,048 training pairs, 1,011 validation pairs, and 2,304 testing pairs.

**Implementation Details.** We use an NVIDIA GeForce RTX 3090 GPU and PyTorch 1.13.1 for training. Image features are downsampled to $\{24 \times 32, 12 \times 16, 6 \times 8\}$. We use level-specific sweep window sizes $o = \{8, 4, 2\}$ across the three levels. For the DLA strategy, we set the maximum iteration steps to 4 and adopt dynamic step selection with an action space of $\{0, 1, 2, 3\}$ to control the interaction depth. We set $\xi_1 = \xi_2 = 1$.

**Metrics.** We evaluate the models using three metrics: Inlier Ratio (IR), which measures the percentage of pixel-to-point matches within 5 cm; Feature Matching Recall (FMR), which indicates the proportion of image–point cloud pairs with an IR greater than 10%; and Registration Recall (RR), which represents the proportion of pairs with an RMSE below 10 cm. T-SNE (Van der Maaten & Hinton, 2008) is a dimensionality reduction technique used to visualize high-dimensional data in 2D or 3D, while Maximum Mean Discrepancy (MMD) (Borgwardt et al., 2006) is a statistical measure that quantifies the difference between two distributions.

## 4.2. Evaluations on Dataset

**Quantitative Comparison.** We compare our method with prior approaches (Choy et al., 2019; Wang et al., 2021; Huang et al., 2021b; Li et al., 2023; Cheng et al., 2025b;c; Mu et al., 2025; An et al., 2025) on RGB-D Scenes V2 and 7-Scenes (Table 1). Overall, our FS-I2P achieves consistent gains across the three evaluation metrics, indicating improved correspondence quality and more reliable pose

estimation under scale ambiguity. On RGB-D Scenes V2, FS-I2P attains the best mean Inlier Ratio of 42.9, outperforming Flow-I2P (40.1) by 2.8 pp and 2D3D-MATR (32.4) by 10.5 pp, which validates that the proposed Focus-Sweep interaction strengthens cross-modal alignment. Although Feature Matching Recall (FMR) on 7-Scenes is already saturated for several methods, FS-I2P still maintains competitive FMR (94.4 on RGB-D Scenes V2 and 92.4 on 7-Scenes). More importantly, FS-I2P yields a clear improvement in Registration Recall (RR): on RGB-D Scenes V2, the mean RR reaches 72.9, surpassing 2D3D-MATR (56.4) by 16.5 pp and Flow-I2P (68.4) by 4.5 pp, with particularly large margins on Scene-12 and Scene-14. On 7-Scenes, FS-I2P significantly improves challenging cases such as Heads (52.1 to 86.3 compared with 2D3D-MATR), where the small median depth amplifies small pose errors, and it also improves Stairs, where repetitive patterns frequently trigger scale-confusing correspondences. Finally, using DINOv2 as a stronger image encoder further boosts performance: FS-I2P+DINOv2 achieves the best RR on both benchmarks (86.3 on RGB-D Scenes V2 and 84.6 on 7-Scenes). This aligns with our motivation that scale ambiguity is exacerbated by repetitive textures and viewpoint changes; DINOv2 provides more discriminative, context-aware features, making the Focus-Sweep interactions more reliable and enabling dynamic depth allocation to better adapt the number of iterations, thereby reducing mismatches.

**Qualitative Comparison.** Figure 4 compares our registration results with the baseline. Correspondences with a reprojection offset below 60 pixels are colored green, while the remaining mismatches are shown in red. The highlighted regions (yellow arrows/boxes) indicate where our method yields clear improvements, producing more correct matches (green) and suppressing incorrect ones (red). In Figure 5, we further validate the estimated pose by projecting the point cloud onto the image using the predicted transformation, where the projected points exhibit only minor offsets and align well with the image content.

**Ablation Study.** Table 2 summarizes the ablation study on RGB-D Scenes V2. Starting from the HTS-only baseline (M1), removing HTS (M2) leads to a consistent degradation across all metrics, confirming its essential role in cross-modal interaction. Building upon HTS, introducing the proposed Sweep module (M3) yields a clear performance improvement (RR: 56.4→62.4), demonstrating that region-wise refinement effectively reduces mismatches. Further

*Table 3.* Comparison of time and memory.

| Module | Convergence (Epoch) | Inference Time (s) | Memory (MB) |
|---|---|---|---|
| Base | **19** | **0.281** | **6240** |
| Transformer | 25 | 0.497 | 8346 |
| FS-I2P$_{(Ours)}$ | 21 | 0.304 | 6415 |

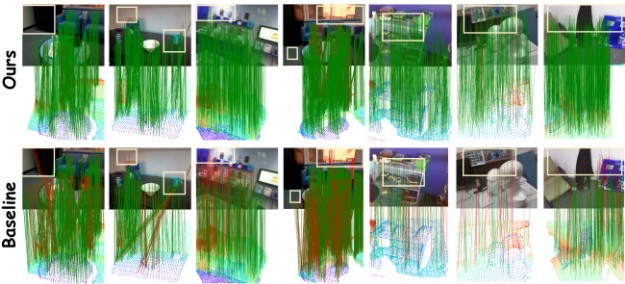

*Figure 4.* Visulization of matching accuracy.

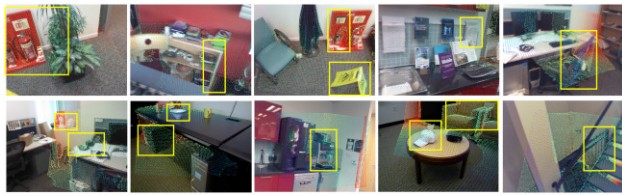

*Figure 5.* Visualization of point cloud projections onto the image.

incorporating the Focus module together with Sweep (M4) results in a substantial gain (RR: 68.2), indicating that global context modeling complements local refinement. In contrast, applying Focus without Sweep (M5) brings only marginal improvement (RR: 59.7), highlighting that iterative refinement is crucial for robust registration. When removing HTS while enabling DLAS (M6), the performance remains limited, suggesting that HTS is still necessary even under adaptive depth allocation. Finally, equipping the full model with Focus, Sweep, HTS, and DLAS (M7) achieves the best performance (IR/FMR/RR: 40.1/94.4/72.9), demonstrating that dynamically allocating interaction depth further enhances robustness and pose accuracy. In addition to effectiveness, Table 3 shows that FS-I2P achieves these gains with only minor computational overhead compared to the base model (0.304 s, 6415 MB vs. 0.281 s, 6240 MB), while remaining significantly more efficient than the transformer-based alternative (0.497 s, 8346 MB). These results highlight a favorable trade-off between performance and efficiency.

## 5. Conclusion

We propose a Hierarchical Focus–Sweep Registration Network with Dynamically Allocated Depth for image-to-point cloud registration, aiming to reduce scale-ambiguity-induced mismatches in challenging scenarios. FS-I2P adopts a cognitively inspired Focus–Sweep interaction built on Mamba: the Focus operation captures global context and coarse correspondences using point-cloud scale cues, while the Sweep operation performs region-wise refinement via repeated image inspection and point-cloud rechecking. Moreover, we introduce a Dynamic Layer Allocation Strategy that uses reinforcement learning with global registration rewards to adaptively determine the required interaction depth. Extensive experiments and ablations on RGB-D Scenes v2 and 7-Scenes demonstrate that FS-I2P achieves state-of-the-art performance.

## Acknowledgements

The authors gratefully acknowledge the support from the Science and Application Research Project of the Tianwen-2 Mission under China's Planetary Exploration Engineering Program, Grant No. TW2-01-001.

## Impact Statement

This paper presents work whose goal is to advance the field of Machine Learning. There are many potential societal consequences of our work, none which we feel must be specifically highlighted here.

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

## A. T-SNE and MMD

**t-SNE.** t-Distributed Stochastic Neighbor Embedding (t-SNE) (Van der Maaten & Hinton, 2008) is a nonlinear dimensionality reduction technique used primarily for data visualization. It is particularly effective for visualizing high-dimensional datasets by embedding them into two or three dimensions. t-SNE aims to preserve the local structure of data points by modeling similar objects with nearby points and dissimilar objects with distant points. This method is beneficial for visualizing multimodal tasks, allowing for intuitive insights into complex datasets that span various domains such as images, text, and audio.

**MMD.** Maximum Mean Discrepancy (MMD) is a statistical method used to compare two probability distributions. It is a non-parametric technique that measures the difference between distributions by mapping data into a high-dimensional feature space using a kernel function. MMD is widely used in various machine learning applications, such as generative adversarial networks (GANs) (Salimans et al., 2016), domain adaptation, and distribution testing. The core idea of MMD is to compute the distance between the means of two distributions in a reproducing kernel Hilbert space (RKHS) (Berlinet & Thomas-Agnan, 2011). Given two distributions $I$ and $Q$, the MMD is defined as:

$$\text{MMD}(I, Q) = \left\| \mathbb{E}_{x \sim I}[\phi(x)] - \mathbb{E}_{y \sim Q}[\phi(y)] \right\|_{\mathcal{H}}, \quad (18)$$

where $\phi$ is the feature mapping function induced by a kernel, and $\mathcal{H}$ is the RKHS.

MMD is applied in various areas, including generative models for evaluating the similarity between the distributions of generated and real data, domain adaptation for reducing distribution shifts between source and target domains, and hypothesis testing for determining if two samples are drawn from the same distribution.

## B. Positional embedding

We augment the 2D and 3D features with their positional information before the attention layer.

$$\hat{F}^{\mathcal{I}}_{\text{pos}} = \hat{F}^{\mathcal{I}} + \phi(\hat{Q}), \quad \hat{F}^{\mathcal{P}}_{\text{pos}} = \hat{F}^{\mathcal{P}} + \phi(\hat{P}). \quad (19)$$

The Fourier embedding function $\phi(x)$ (Mildenhall et al., 2021) encodes positional information by transforming it into a sequence of sine and cosine terms:

$$\phi(x) = \big[ x, \sin(2^0 x), \cos(2^0 x), \dots, \\ \sin(2^{L-1} x), \cos(2^{L-1} x) \big], \quad (20)$$

where $L$ is the length of the embedding. This transformation incorporates spatial positioning into the features. To facilitate further computations, the first two spatial dimensions of the 2D features are flattened, making the augmented features $\hat{F}^{\mathcal{I}}_{\text{pos}}$ and $\hat{F}^{\mathcal{P}}_{\text{pos}}$ ready for subsequent processing.

## C. Network architecture

With the rapid advancement of deep learning (Wang et al., 2023a;b; 2025a; Xie et al., 2021; Zha et al., 2025a; Tao et al., 2026; Wang et al., 2025b; 2024a;c; 2022; 2023e; Zhang et al., 2022; Zha et al., 2025b; Wang et al., 2023d; Cheng et al., 2026a;b; Pan et al., 2024a;b; 2025; Cheng et al., 2024; Jin et al., 2025), we utilize a 4-stage ResNet (He et al., 2016) with a Feature Pyramid Network (FPN) (Lin et al., 2017) as the image backbone. The output channels for each stage are $\{128, 128, 256, 512\}$. The input images have a resolution of $480 \times 640$ pixels, which is downsampled to $60 \times 80$ in the coarsest level for efficiency. For the 3D backbone, a 4-stage KPFCNN (Thomas et al., 2019) is employed, with output channels configured as $\{128, 256, 512, 1024\}$. Point clouds are voxelized with an initial voxel size of 2.5 cm, which doubles at each stage.

At the coarse level, 2D features are resized to $24 \times 32$ pixels before being fed into the transformer to enhance computational efficiency. Each transformer layer has 256 feature channels, 4 attention heads, and uses ReLU activation functions. In the patch pyramid setup, the coarsest level begins with $H_0 = 6$ and $W_0 = 8$, expanding through 3 pyramid levels: $\{6 \times 8, 12 \times 16, 24 \times 32\}$. At the fine level, both 2D and 3D features are projected into a 128-dimensional space for feature matching.

To address significant misalignments caused by structurally similar but non-overlapping regions, we incorporate ground-truth supervision during training by evaluating the overlap ratio between the annotated image-point cloud correspondences, using the dataset-provided ground-truth pose. This overlap measure is computed based on the local neighborhoods of image keypoints and point cloud nodes, which are projected according to the ground-truth pose. However, relying solely on this overlap criterion results in sparse and discontinuous reward signals, as only a limited number of correspondences exhibit valid overlaps, leaving most candidate pairs unlabeled. We incorporate the overlap signal in combination with the rotation-invariant geometric similarity to form the final reward signal for policy optimization. This addition helps prevent large misalignments by providing supervisory information when available, without significantly influencing the reinforcement learning process in regions lacking explicit annotations. As such, it ensures the network benefits from strong supervision in well-annotated areas, while still receiving dense and consistent feedback in less annotated regions, thereby supporting stable learning.

We define ground truth using bilateral overlap (Huang et al.,

*Table 4.* Ablation results on the number of Transformer layers and Mlayers.

| Setting | # Layer Number | | | | |
|---|---|---|---|---|---|
| | 1 | 2 | 3 | 4 | 5 |
| Transformer | 49.8 | 53.5 | **56.4** | 54.3 | 51.5 |
| Setting | # Layer Number | | | | |
| | 0 | 1 | 2 | 3 | 4 |
| FS layer | 56.4 | 57.1 | 60.4 | **65.3** | 62.5 |

*Table 6.* Evaluation Results on KITTI Dataset.

| Method | Type | RTE(m) ↓ | RRE (°) ↓ |
|---|---|---|---|
| vpc + GeoTransformer(Qin et al., 2023) | Point-to-Point | 4.27 ± 7.14 | 8.67 ± 8.55 |
| vpc + Hunter | Point-to-Point | 4.59 ± 5.22 | 6.23 ± 5.17 |
| DeepI2P(2D) (Li & Lee, 2021) | Image-to-Point | 5.15 ± 7.35 | 9.14 ± 8.02 |
| CorrI2P (Ren et al., 2022) | Image-to-Point | 4.24 ± 7.26 | 6.47 ± 5.20 |
| VP2P (Yue et al., 2025) | Image-to-Point | 2.05 ± 3.23 | 4.01 ± 6.37 |
| 2D3D-MATR (Li et al., 2023) | Image-to-Point | 1.86 ± 3.79 | 2.59 ± 4.46 |
| RetrI2P (Bie et al., 2025) | Image-to-Point | 1.61 ± 2.39 | 3.16 ± 2.85 |
| FreeReg (Wang et al., 2023c) | Image-to-Point | 1.78 ± 1.76 | 2.89 ± 4.47 |
| CFI2P (Yao et al., 2024) | Image-to-Point | 1.95 ± 2.97 | 2.63 ± 3.19 |
| FS-I2P (Ours) | Image-to-Point | **1.57 ± 2.75** | **2.36 ± 2.58** |

2021a). A patch pair is positive if both the 2D and 3D overlap ratios are at least 30%, and negative if both are below 20%. The 2D and 3D overlap ratios are used as $\lambda_p$, while $\lambda_n$ is set to 1. At the fine level, a pixel-point pair is positive if the 3D distance is below 3.75 cm and the 2D distance is under 8 pixels, and negative if the 3D distance exceeds 10 cm or the 2D distance is over 12 pixels. Scaling factors are set to 1. Pairs not meeting these criteria are ignored as the safe region during training. The margins are set to $\Delta_p = 0.1$ and $\Delta_n = 1.4$.

## D. Addtional ablation study

Table 4 studies the effect of interaction depth. For the Transformer baseline, increasing the number of layers improves RR from 49.8 to 56.4 when going from 1 to 3 layers, but deeper stacking causes performance to drop (54.3 at 4 layers and 51.5 at 5 layers), indicating over-smoothing and amplified early attention bias. In contrast, the proposed Focus-Sweep (FS) interaction consistently benefits from additional iterations up to a moderate depth: starting from 0 FS layers (56.4), RR increases to 57.1/60.4/65.3 for 1/2/3 layers, and then slightly decreases at 4 layers (62.5). These results suggest that iterative Focus-Sweep refinement is more depth-efficient and stable than simply stacking Transformer layers, and that a moderate number of FS iterations provides the best trade-off.

Better features can have a crucial impact during multi-scale interactions, because superior feature representations are more accurate and make it easier to discriminate across different scales. Table 5 compares our method with the baseline when both incorporate DINOv2 features. FS-I2P+DINOv2

*Table 5.* Comparison between our method and the baseline after incorporating DINOv2 features.

| Method | FMR | IR | RR |
|---|---|---|---|
| Baseline+Dinov2 | 37.8 | 92.4 | 74.2 |
| FS-I2P+Dinov2 | **41.2** | **94.5** | **86.3** |

consistently outperforms Baseline+DINOv2 across all metrics, improving FMR from 37.8 to 41.2 and IR from 92.4 to 94.5. More importantly, it achieves a large gain in RR (74.2 to 86.3), indicating that our Focus-Sweep interactions convert stronger image semantics into more reliable correspondences and substantially better pose estimation, rather than merely increasing matching scores.

Although FS-I2P is primarily designed for indoor image-to-point cloud registration, we further evaluate it on the outdoor KITTI benchmark to examine generalization (Table 6). FS-I2P achieves the best performance among all compared methods, obtaining 1.57±2.75 m RTE and 2.36±2.58° RRE. This result surpasses recent Image-to-Point approaches such as 2D3D-MATR (1.86±3.79 m, 2.59±4.46°) and RetrI2P (1.61±2.39 m, 3.16±2.85°), demonstrating that the proposed Focus-Sweep interaction and dynamic iteration mechanism are not restricted to indoor scenes and can effectively handle the scale variation and viewpoint changes in outdoor driving scenarios.

Figure 6 provides a layer-wise visualization of the model's focus, showing a clear progression from broad, dense coverage to sparse, concentrated attention. In early layers, the model activates over large portions of the image and point cloud, indicating a global scan that captures overall context and coarse alignment cues. As depth increases, the focus becomes more selective and shifts toward structurally informative regions such as boundaries, corners, and distinctive layouts, while suppressing flat or repetitive areas that are prone to scale ambiguity and mismatches. This consis-

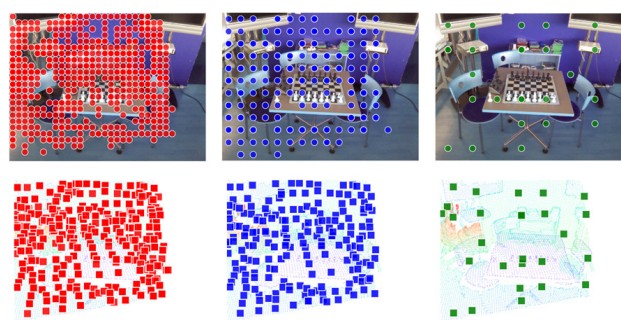

*Figure 6.* Layer-wise visualization of the model's focus.

tent convergence on both modalities suggests that iterative cross-modal interaction gradually filters unreliable correspondences and retains high-value regions that contribute more effectively to accurate pose estimation.

## E. Rebuttle

### E.1. Comparison with Transformer

We design a fully consistent sweep-focus framework, where the Mamba layers are replaced with a standard Transformer and a Swin Transformer. All experiments are conducted on RGB-D V2 with a fixed depth of two layers.

*Table 7.* Comparison with Transformer-based variants.

| Method | RR | Memory | Time |
|---|---|---|---|
| Mamba | **69.9** | **6327** | **0.312** |
| Transformer | 65.8 | 8134 | 0.502 |
| Swin-Transformer | 69.7 | 7932 | 0.481 |

These results provide several important insights. First, compared with the standard Transformer, Mamba achieves clear gains in both efficiency and performance, reducing memory usage and inference time while also improving RR. This indicates that the improvements are closely related to the SSM-based sequence modeling mechanism, which enables efficient long-range dependency modeling without quadratic complexity. Second, compared with Swin-Transformer, which introduces window-based local interactions, Mamba achieves slightly better performance (69.9 vs. 69.7) while maintaining significantly lower computational cost. Notably, the sweep-focus framework itself is naturally compatible with window-style interaction patterns, as the sequential scanning and aggregation process aligns well with localized token grouping. This explains why Swin-Transformer performs competitively in this framework. However, Mamba still provides both higher accuracy and better efficiency, demonstrating a more favorable overall trade-off. Overall, under controlled settings, these results demonstrate that the proposed SSM-based Sweep module offers a more efficient alternative to cross-attention-based designs, while maintaining strong performance. This supports our claim that the improvements stem from the intrinsic properties of the SSM mechanism, rather than other architectural factors.

### E.2. Analysis of Dynamic Layer Allocation Policy

We further analyze the choice of layer numbers under different mean depths. For clarity, we use *scale* to denote the number of FS-layers. The results indicate that, among the static configurations, a fixed interaction depth of 2 achieves the best performance. Performance improves as the depth increases from 1 to 2, but degrades at depth 3, suggest-

*Table 8.* Performance under different fixed depths.

| Scale | RR |
|---|---|
| 0 | 56.4 |
| 1 | 63.2 |
| 2 | 68.2 |
| 3 | 67.7 |
| Dynamic (Ours) | **72.9** |

ing that excessive interaction may introduce noise and hinder learning. In contrast, the proposed dynamic strategy further enhances performance, achieving the best result of 72.9 and outperforming all fixed-depth variants. This demonstrates that different samples benefit from different interaction depths, highlighting the advantage of adaptively adjusting the interaction depth for each sample.

*Table 9.* Distribution of selected depths under different scales (%).

| | Scale=0 | Scale=1 | Scale=2 | Scale=3 |
|---|---|---|---|---|
| Depth1 (1.74) | 16.50 | **24.33** | 31.83 | 27.33 |
| Depth2 (1.66) | 19.50 | 22.33 | 30.33 | **27.83** |
| Depth3 (1.18) | 17.83 | 22.67 | **34.50** | 25.00 |
| Depth4 (1.39) | **21.83** | 21.17 | 31.83 | 25.17 |

The learned depth distributions are well balanced, with all depth levels being utilized. In particular, depth=2 is selected most frequently, while depth=0 also maintains a non-negligible proportion, indicating that the policy learns to skip unnecessary interactions for simpler cases. This demonstrates that the dynamic allocation does not collapse to a single depth, but instead adapts to sample difficulty.

### E.3. Sequential Token Design

Our design mainly exploits the interactive properties of Mamba to refine the interaction process. We further evaluate different token ordering strategies. *RasterOrder* arranges tokens in a row-wise manner from left to right and top to bottom. *ReverseOrder* follows the opposite direction, from bottom to top and right to left. *Column-wise* arranges tokens column by column, from top to bottom and left to right. We

*Table 10.* Comparison of different token ordering strategies.

| Strategy | IR ↑ |
|---|---|
| Raster Order | 38.8 |
| Reverse Order | 35.8 |
| Column-wise | 37.5 |
| Ours | **40.1** |

use IR as the evaluation metric since it directly reflects the quality of cross-modal matching and retrieval, which is most relevant to the effectiveness of the sequential token design. The results show that our design achieves the best performance among all ordering strategies. This demonstrates that properly leveraging the sequential modeling capability of Mamba can further enhance cross-modal interaction effectiveness.

### E.4. Generalization Across Scenes with Varying Complexity

We further evaluate the generalization ability of the proposed method across scenes with varying complexity and scale, including more challenging outdoor scenarios. The results demonstrate that our method maintains competitive performance under more challenging and diverse scenarios, indicating strong generalization capability across scenes with varying complexity and scale.

*Table 11.* Generalization performance across different scenes (7-Scenes→RGB-DScenesv2 ).

| Model | IR ↑ | FMR ↑ | RR ↑ |
|---|---|---|---|
| 2D3D-MATR | 17.5 | 66.2 | 19.8 |
| FS-I2P | **19.2** | **68.6** | **24.4** |

### E.5. Discrete Action Space Analysis

We agree that Gumbel-Softmax is a viable alternative for discrete decision modeling. However, since our action involves selecting a discrete number of interaction steps, REINFORCE provides a more direct and interpretable formulation without introducing approximation bias. We further conduct comparative experiments, and the results are summarized in Table 12. The results clearly show that REINFORCE achieves significantly better performance than Gumbel-Softmax, demonstrating its effectiveness for our discrete interaction selection strategy.

*Table 12.* Comparison of different strategies for discrete action modeling.

| Method | RR ↑ |
|---|---|
| Gumbel-Softmax | 66.4 |
| Reinforce | **72.9** |

### E.6. Stability of RL-based Training

We thank the reviewer for this insightful question. In practice, we observe that our RL-based optimization is stable and not highly sensitive to hyperparameters or initialization. This is mainly due to the lightweight design of the policy

network and the simple reward formulation.

**(1) Reward Scaling ($\delta$).** We evaluate different values of the reward scaling factor $\delta$, including $1 \times 10^{-5}$, $1 \times 10^{-6}$ (default), and $1 \times 10^{-7}$. The results are shown in Table 13. All settings lead to stable training and comparable performance, with only marginal differences. This indicates that the training process is robust to the choice of reward scaling.

*Table 13.* Effect of reward scaling factor $\delta$.

| $\delta$ | IR ↑ |
|---|---|
| $1 \times 10^{-5}$ | 39.9 |
| $1 \times 10^{-6}$ | **40.1** |
| $1 \times 10^{-7}$ | 39.8 |

**(2) Sensitivity to Policy Initialization.** We further investigate different initialization strategies for the policy network. In addition to random initialization, we also adopt a similarity-based top-$k$ selection as initialization. The results are summarized in Table 14. The results are highly consistent across different settings, suggesting that the learned policy does not strongly depend on initialization. We also observe that similarity-based initialization slightly improves convergence speed, and we will incorporate this into the revised manuscript.

*Table 14.* Sensitivity to policy initialization.

| Setting | RR ↑ | Convergence ↓ |
|---|---|---|
| None | 72.9 | 21 |
| Cosine | **73.1** | **20** |

**(3) Mode Collapse.** We acknowledge that mode collapse was observed in early experiments. When the candidate depths were limited to $\{1, 2, 3\}$, the policy tended to favor the maximum depth (i.e., 3) during the initial training stage. This is likely because deeper interactions often yield larger immediate rewards before the policy is well calibrated, leading to a biased reward distribution. To address this issue, we introduce depth $= 0$ as an additional action, which represents skipping interaction. This provides a true baseline option without additional refinement. As a result, the policy no longer needs to choose only among progressively deeper interactions, but can adaptively skip unnecessary steps for simpler samples. This modification reshapes the reward landscape, alleviates the bias toward maximum depth, and encourages more balanced exploration across different depths. Consequently, the mode collapse issue is effectively mitigated.

### E.7. Outdoor Scenario Adaptation

We agree that outdoor scenes differ substantially from indoor ones in terms of spatial scale and overlap ratio, which makes the Focus module more challenging to apply directly. To address this issue, we introduce an additional learnable parameter that is multiplied with $\alpha$, $\beta$, and $\gamma$, allowing the global normalization process to adapt dynamically across different outdoor scenes. This design effectively mitigates the bias caused by large-scale spatial mismatch between the point cloud and the image field of view, and improves the stability of the model in outdoor environments.

### E.8. Mamba Token Ordering

We agree that token ordering is an important factor for SSM-based models. To examine its impact in our Sweep phase, we conduct an additional experiment using bidirectional scanning as a representative alternative ordering strategy. This setting is commonly adopted to reduce directional bias while maintaining a relatively simple sequential structure.

*Table 15.* Effect of token ordering strategies.

| Strategy | IR ↑ | FMR ↑ | RR ↑ |
|---|---|---|---|
| Bidirectional Scan | 38.7 | 92.6 | 70.1 |
| Ours | **40.1** | **93.3** | **72.9** |

As shown in Table 15, our ordering strategy achieves the best performance across all metrics. We attribute this improvement to the fact that the proposed Sweep design preserves a more coherent spatial progression and cross-modal interaction pattern, which better aligns with the sweep-focus mechanism. In contrast, bidirectional scanning may disrupt interaction continuity and lead to less stable token transitions.

