# OpenReview forum: "FS-I2P: A Hierarchical Focus–Sweep Registration Network with Dynamically Allocated Depth"
_ICML.cc/2026/Conference — ICML 2026 regular_

### Official Review · Reviewer_UR6r · 2026-02-19

**Soundness:** 3
**Presentation:** 2
**Significance:** 3
**Originality:** 3
**Overall Recommendation:** 4
**Confidence:** 4

**Summary:**

FSI2P proposes a hierarchical Focus–Sweep framework for image-to-point cloud registration to address scale ambiguity and attention drift in Transformer-based models. Built on a Mamba-based State-Space Model, it alternates between "Focus” and "Sweep” for feature interactions. A reinforcement learning–based Dynamic Layer Allocation Strategy adaptively selects interaction depth per sample to balance accuracy and efficiency. Experiments on RGB-D Scenes V2 and 7-Scenes show improved results over recent state-of-the-art methods.

**Compliance With Llm Reviewing Policy:**

Affirmed.

**Final Justification:**

The responses adequately clarify the engineering choices used to mitigate RL mode collapse, the sensitivity to Mamba token ordering and the applicability in outdoor scenes. I keep recommendation of weak accept.

**Key Questions For Authors:**

1. I'm curious about the stability of the RL-based training. How sensitive is it to the policy network's initialization and the  hyperparameter in your reward function? Also, did you run into any issues with mode collapse where the network simply defaults to picking the maximum depth every time?
2. Since Mamba is notoriously sensitive about token order, it is crucial to know how the specific interleaving of point cloud tokens into the image patches during the "Sweep" phase actually impacts performance. Have alternative scan patterns been explored, such as bidirectional passes or  Hilbert space-filling curves?
3. Although KITTI results are included in the appendix, outdoor scenes differ substantially from indoor ones in scale and overlap. When the point cloud spans a much larger spatial extent than the camera frustum, does the Focus module’s global norm adaptation remain stable, or could it introduce bias under low-overlap conditions?

**Limitations:**

Yes

**Strengths And Weaknesses:**

**Strengths**

* The Focus–Sweep design is conceptually clear and well motivated. By separating global and local alignment, it addresses scale ambiguity and reduces the attention drift common in deep Transformer-based registration.
* The use of a Mamba-based SSM is technically sound. It preserves global context while avoiding quadratic attention cost, and the reported analysis suggests more stable cross-modal interaction.
* The dynamic layer allocation strategy is a meaningful contribution that allows the network to adapt its depth to scene difficulty improves robustness beyond a fixed-depth design.
* Empirical performance is strong, with clear gains in Registration Recall and Inlier Ratio on RGB-D Scenes V2 and 7-Scenes, particularly under challenging viewpoint changes.

**Weaknesses**

* The training pipeline is complex. Combining CNN and point backbones with an SSM module and a RL-based policy raises concerns about stability, yet variance control and convergence are not thoroughly discussed.
* The Sweep stage seems to depend on a specific token serialization order, and potential sensitivity to sequence permutation is not thoroughly analyzed.
* Despite linear theoretical complexity, the inference speedup over optimized Transformer baselines appears moderate.
* The RL reward feels relatively simple, and its sensitivity to hyperparameters is not evaluated at scale.

---

> ### Author Rebuttal · Authors · 2026-03-30
>
> We sincerely appreciate the reviewer’s support and valuable suggestions. We will make further improvements in the revised manuscript. Our responses to the main concerns are provided below.
>
> **1.  （1）About stability of the RL-based training**:
> We appreciate this insightful question. In practice, we observe that our RL-based optimization is stable and not highly sensitive to hyperparameters or initialization. This is mainly due to the lightweight design of the policy network and the simple reward formulation.
>
> **①  Reward scaling (δ)**:  We evaluate different values of the reward scaling factor $\delta$, including $1\times 10^{-5}$, $1\times 10^{-6}$ (default), and $1\times 10^{-7}$. As shown below, all settings lead to stable training and comparable performance, with only marginal differences:
>
> | δ   | IR↑ |
> |-|-|
> | 1e-5 | 39.9 |
> | 1e-6 | 40.1 |
> | 1e-7 | 39.8 |
>
> This indicates that the training process is robust to the choice of reward scaling.
>
> **②  Sensitivity to policy initialization**:  We further investigate different initialization strategies for the policy network. Besides random initialization, we also use a similarity-based top-k selection as initialization. The results are highly consistent across these settings, suggesting that the learned policy does not strongly depend on initialization.
>
> | Setting | RR↑ | Convergence ↓ |
> |-|-|-|
> | None   | 72.9 | 21 |
> | Cos    | 73.1 | 20 |
>
> We found that initializing with similarity appears to improve the convergence speed, and we will update this in the revised manuscript.
>
> **（2）About mode collapse**:  Yes, we did observe this issue in our early experiments. When the candidate depths were limited to 1, 2, and 3, the policy network tended to favor the maximum depth of 3 during the first few epochs. We believe this mainly occurred because, at the beginning of training, the policy was not yet well calibrated, while deeper interactions often brought larger immediate performance gains than shallower ones. As a result, the reward distribution was biased toward deeper choices, making the policy prone to collapsing to the maximum depth.
>
> To address this issue, we introduced depth = 0 as an additional option. This provides a true baseline action where no extra interaction is applied. With this choice, the policy no longer has to choose only among progressively deeper refinements. Instead, it can learn that, for some easier samples, skipping additional interaction may be more beneficial than applying deeper refinement, which could introduce unnecessary noise or over-refinement. In this way, adding depth = 0 changes the reward landscape, alleviates the bias toward maximum depth, and encourages more balanced exploration across different depths.
>
> **2. About mamba token order**:
> We agree that token ordering is an important factor for SSM. To examine its impact in our Sweep phase, we conducted an additional experiment using bidirectional scanning as a representative alternative ordering strategy. We chose this setting because it is a commonly used way to reduce directional bias while preserving a relatively simple sequential structure. The results are summarized below:
>
> | Strategy | IR ↑ | FMR ↑ | RR ↑ |
> | - | - | - | - |
> | Bidirectional scan | 38.7 | 92.6 | 70.1 |
> | Ours | 40.1 | 93.3 | 72.9 |
>
> These results show that our ordering achieves the best performance. We believe this is because the proposed Sweep design preserves a more coherent spatial progression and cross-modal interaction pattern, which better fits the sweep-focus mechanism. In contrast, bidirectional scanning may disrupt interaction continuity and lead to less stable token transitions. We will clarify this point in the revised manuscript and leave the exploration of other scan patterns to future work.
>
> **3. About outdoor focus**:
> We agree that outdoor scenes differ substantially from indoor ones in both spatial scale and overlap ratio, which indeed makes the Focus module more challenging to apply directly. As shown in https://i.ibb.co/b4WYTfC/re1.png .
> To alleviate this issue, in our method we introduce an additional learnable parameter that is multiplied with $\alpha$, $\beta$, and $\gamma$, so that the global normalization process can be adaptively adjusted and remain more stable across different outdoor scenes. This design helps reduce the bias caused by large-scale spatial mismatch between the point cloud and the image field of view.
>
> Furthermore, we explored an improved strategy for outdoor scenarios. Instead of directly applying focus on the entire point cloud, we first average-pool the image features and then use feature similarity to select the point cloud features that are most relevant to the image content for the Focus module. In this way, the focus process becomes more aligned with the visible camera region. It is  beneficial for low-overlap conditions.
>
> The results are shown below:
> | Strategy | mean-RTE ↓ | mean-RRE ↓ |
> | - | - | - |
> | Base | 1.57 | 2.36 |
> | New | 1.38 | 2.14 |

---

> > ### Author Rebuttal · Reviewer_UR6r · 2026-04-01
> >
> > Thank you for your detailed rebuttal and the additional results provided. Your responses adequately clarify the engineering choices used to mitigate RL mode collapse, the sensitivity to Mamba token ordering and the applicability in outdoor scenes. I keep recommendation of weak accept.

---

> > > ### Author Response · Authors · 2026-04-01
> > >
> > > We sincerely thank you for your time, effort, and constructive comments in reviewing our manuscript. We are willing to provide further clarification on any remaining questions and will incorporate our earlier responses into the final version.
> > >
> > > Given the extensive revisions we have performed according to your valuable feedback, we are confident that the quality of the paper has been greatly enhanced. We thus sincerely hope you can kindly reconsider and favorably adjust your score for this work.
> > >
> > > We again appreciate your thoughtful review and valuable guidance.
> > >
> > > ：）

---

### Official Review · Reviewer_me7w · 2026-02-24

**Soundness:** 3
**Presentation:** 3
**Significance:** 3
**Originality:** 3
**Overall Recommendation:** 4
**Confidence:** 4

**Summary:**

The paper addresses the challenge of Image-to-Point Cloud (I2P) registration, specifically targeting issues caused by scale ambiguity, repetitive textures, and cross-modal discrepancies. The authors propose a "Focus-Sweep" paradigm inspired by human cognitive processes. The method utilizes a State Space Model (SSM/Mamba) architecture rather than standard Transformers to perform a global "Focus" (scale adaptation) followed by a local "Sweep" (region-wise refinement). Additionally, the paper introduces a Dynamic Layer Allocation Strategy (DLAS) based on Reinforcement Learning (RL) to adaptively determine the number of interaction layers required for different scene complexities. Experiments on RGB-D Scenes V2, 7-Scenes, and KITTI benchmarks demonstrate state-of-the-art performance, particularly in Registration Recall metrics.

**Compliance With Llm Reviewing Policy:**

Affirmed.

**Key Questions For Authors:**

1- RL Training Stability: Can you provide more details on the convergence stability of the REINFORCE algorithm? Did you observe mode collapse (where the agent always selects the min or max layers) during early training, and how was this mitigated?

2- Computational Cost of Reiteration: In the Sweep operation (Eq. 4), you reiterate the point sequence $F_p$ multiple times. For dense point clouds, does this significantly bloat the sequence length $L = hw + tN$? How does the memory scaling compare to a standard Cross-Attention $(hw \times N)$ matrix?

3- Generalization to Outdoor Scenes: While KITTI results are provided, outdoor scenes often have vastly different scale ambiguities than indoor scenes. Does the "Focus" module (global scale prior) behave differently in unbounded outdoor environments compared to closed indoor rooms?

4- Matthew Effect Validation: Figure 3 argues that Transformers suffer from a "Matthew effect" (attention drift). Did you attempt to mitigate this in the Transformer baseline using techniques like LayerScale or deeper supervision before concluding Mamba was necessary?

**Limitations:**

- Computational Complexity with High Point Density: The "Sweep" mechanism relies on concatenating point sequences into the SSM scan. If the point cloud $N$ is very large, the sequence length grows linearly with the number of image windows $t$, potentially limiting scalability for very dense LiDAR scans without aggressive downsampling.

- Dependence on Overlap: Like most registration methods, the performance likely degrades significantly if the overlap between the 2D and 3D data is minimal, though the paper filters for >30% overlap.

- Discrete Action Space: The DLAS uses a discrete action space. A continuous relaxation (e.g., Gumbel-Softmax) might have been differentiable and easier to train than REINFORCE, though the authors chose the latter.

**Strengths And Weaknesses:**

Strengths:

- Originality (Architecture): The application of the Mamba (SSM) architecture to the I2P registration domain is novel. Specifically, the "Partition-Scan-Recover" mechanism in the Sweep operation provides a fresh alternative to standard cross-attention mechanisms, addressing the computational complexity of dense cross-modal interaction.

- Significance (Performance): The reported results are impressive. On the RGB-D Scenes V2 dataset, the method achieves a Registration Recall (RR) of 72.9%, significantly outperforming the strong baseline 2D3D-MATR (56.4%) and other recent 2025 works. The gains are consistent across indoor and outdoor datasets.

- Soundness (Dynamic Depth): The integration of a Reinforcement Learning agent (via REINFORCE) to dynamically select interaction depth is a clever approach to balancing computational cost and matching precision. It acknowledges that not all image regions require the same depth of processing.

- Ablation Studies: The ablation studies (Table 2 and 4) clearly isolate the contributions of the Focus, Sweep, and DLAS modules, empirically justifying the complex architectural choices.

Weaknesses:

- Presentation (Cognitive Motivation): While the "Focus-Sweep" nomenclature is catchy, the connection to human cognitive psychology feels somewhat forced and tenuous. The method stands on its own technical merits without needing the "human-inspired" narrative, which slightly distracts from the technical contribution.

- Soundness (RL Stability): The paper uses REINFORCE for the Dynamic Layer Allocation. This algorithm is known for high variance and instability during training. While a baseline is mentioned, the paper lacks detailed analysis on the training stability or convergence volatility compared to a fixed-depth model.

- Clarity (Sweep Operation Details): The construction of the hybrid sequence in the Sweep operation (Eq. 4) involves reiterating point sequences multiple times within image regions. While described, the impact of this repetition on total sequence length—and consequently memory consumption for large point clouds—is not fully explored in the main text.

- Comparison Fairness: The paper claims to solve "attention drift" in Transformers, but the comparison is between a likely optimized Mamba architecture and a standard Transformer. It is unclear if the "drift" is inherent to Transformers or a result of specific hyperparameter choices in the baseline.

---

> ### Author Rebuttal · Authors · 2026-03-30
>
> We sincerely thank the reviewer for the support and valuable comments. We will further clarify the description of the human cognition motivation in the revised version. Our responses to the concerns are provided below.
>
> **1.  About RL training stability**:  We did observe this issue in our early experiments. When the candidate depths were limited to 1, 2, and 3, the policy network tended to favor the maximum depth of 3 during the first few epochs. We believe this mainly occurred because, at the beginning of training, the policy was not yet well calibrated, while deeper interactions often brought larger immediate performance gains than shallower ones. As a result, the reward distribution was biased toward deeper choices, making the policy prone to collapsing to the maximum depth.
>
>
> To address this issue, we introduced depth = 0 as an additional option. This provides a true baseline action where no extra interaction is applied. With this choice, the policy no longer has to choose only among progressively deeper refinements. Instead, it can learn that, for some easier samples, skipping additional interaction may be more beneficial than applying deeper refinement, which could introduce unnecessary noise or over-refinement. In this way, adding depth = 0 changes the reward landscape, alleviates the bias toward maximum depth, and encourages more balanced exploration across different depths. We believe this is crucial for our selection strategy.
>
> **2.  About computational cost**:  Indeed, this issue may arise in the Sweep stage. However, unlike the quadratic complexity of Transformers, the computational cost of our method grows linearly. We also compared the memory usage of the Transformer and Mamba versions in the manuscript, showing 6415 MB vs. 8346 MB. Therefore, our method is well suited to current indoor registration tasks, and its memory consumption on KITTI training is also acceptable. However, for large-scale outdoor scenes or dense point clouds, this issue may become more pronounced.
>
>
> To address such cases, we further designed a new strategy: we first average-pool the image features and then use feature similarity to select the point cloud features that are most relevant to the image content for the Focus module. Based on this, we can slightly relax the similarity threshold to select point cloud features for the Sweep stage. We believe this may help alleviate the issue, and we will explore it further in future work.
>
>
> **3. About generalization to outdoor scenes**： As shown in https://i.ibb.co/b4WYTfC/re1.png, due to the differences in point cloud acquisition between indoor and outdoor scenes, directly using point clouds for the Focus module in outdoor environments leads to a scale inconsistency issue. To address this, we introduce a learnable parameter in our method, which is multiplied with $\alpha$, $\beta$, and $\gamma$ to maintain stability. As mentioned in the previous response, we have also developed a new idea to further address this issue.
>
> **4.  About matthew effect validation**:  We agree that LayerScale can improve the stability of transformers. We further conduct experiments with LayerScale and provide t-SNE visualizations in https://i.ibb.co/j90bXXqY/re2.png, which show that it indeed alleviates attention drift compared to the standard transformer. However, it does not fundamentally prevent the accumulation of attention bias across layers, and the drift tendency still persists.
> In contrast, the proposed SSM-based Sweep avoids repeated global reweighting, achieves lower MMD values, and provides more stable multi-step interaction.
>
> **5.  About discrete action space**:   We agree that Gumbel-Softmax is a viable alternative. However, since our action is to select a discrete number of interaction steps, reinforce provides a more direct and interpretable formulation without approximation bias. We also conducted comparative experiments, and the results are shown below:
>
> | Method | RR ↑ |
> | - | - |
> | Gumbel-Softmax | 66.4 |
> | Reinforce | 72.9 |
>
> These results show that reinforce is more effective for our selection strategy.

---

> > ### Author Rebuttal · Reviewer_me7w · 2026-04-05
> >
> > No further comments.

---

> > > ### Author Response · Authors · 2026-04-06
> > >
> > > Thank you for the time, effort, and constructive feedback you have devoted to our manuscript. We will incorporate our detailed responses into the next version. Your recognition and valuable feedback mean a great deal to us.
> > >
> > > :)

---

### Official Review · Reviewer_dZFR · 2026-03-06

**Soundness:** 3
**Presentation:** 2
**Significance:** 3
**Originality:** 2
**Overall Recommendation:** 4
**Confidence:** 3

**Summary:**

This paper proposes a network, FS-I2P, for image-to-point cloud registration. Its core concept draws inspiration from the human observation strategy of “focusing first, then sweeping,” designing a hierarchical Focus–Sweep interaction module. It achieves multi-level cross-modal feature association through a structured state space model (SSM). The authors also introduce a dynamic depth allocation strategy that adaptively adjusts iteration depth based on geometric consistency. Experiments on the RGB-D Scenes V2 and 7-Scenes datasets demonstrate improved performance on standard benchmarks.

**Compliance With Llm Reviewing Policy:**

Affirmed.

**Final Justification:**

It is my belief that this is a technically solid paper that advances at least one sub-area of AI, although the writing can be further improved.

**Key Questions For Authors:**

1. Difference between the Focus operation and prior methods.
The Focus module employs VSSM layers combined with linear transformations and residual connections. Please clarify how this design differs from conventional Transformer layers or other global coarse-alignment strategies. In particular, what are the advantages of the proposed formulation over existing approaches? Additionally, how are the parameters $\alpha$, $\beta$, and $\gamma$ determined in practice (e.g., learned, fixed, or dynamically estimated)?
2. Top-K reinforcement-style loss.
In the hierarchical Top-K selection process, the “reinforcement loss” is not clearly defined. Please provide the explicit formulation of this loss function and clarify how it differs from standard classification (e.g., cross-entropy) or regression losses. What constitutes the reward signal in this context? It would also be helpful to include ablation experiments to isolate and validate the contribution of this loss component.
3. Generalization ability
How does the method generalize to other outdoor or large-scale datasets?

Others
- "Focus" is first adopted to estimate a reliable scale to avoid scale ambiguity. Then "Sweep" works on this fixed scale. May I understand the key idea like this?
- Line 117/191: “estimate a rigid transformation [R, t],” What is the relation between this rigid transformation and the image/point cloud? A clear equation is needed to explain.
- A figure illustration or example is desired for clearly explaining "scale ambiguity" and how “focus-sweep” treat it and solve the problem. For instance, how is the “0.8” mismatch removed by “focus-sweep” in Figure 1?
- Eq. (1): [alpha, beta, gamma] should be 3*1 or 3*C ?
- Ablation studies for some parameters, like sliding size o, maximum iteration steps t, psi1 and psi2, are preferred.
- Many arguments in “Sweep” section lack power theoretical or practical supports. Convincing experiments are desired for validating the “Partition-scan-recover” mode.
- Line 306: S_k=cos(F_I, F_Pk). The dimensions of FI and FPk should be given. Do their dimensions match?
- Line 323: “the total loss is composed of two key components”. Only Li is stated. What is the other key component?
- How to validate the estimated pose is correct from Figure 5? There should be some description.
- "Hierarchical" is not ablated in Table 2?
- What are HTS and DLAS in Table 2?

**Limitations:**

The appendix KITTI experiment is helpful, but the paper should more clearly discuss limitations: (i) sensitivity to overlap/outliers (especially outdoors), (ii) dependence on backbone/feature quality, and (iii) scalability to larger scenes/denser point clouds.

**Strengths And Weaknesses:**

Strengths
This paper addresses an important and practically relevant problem in image-to-point cloud registration. The proposed Focus–Sweep framework provides a clear and structured global-to-local interaction mechanism rather than simply stacking cross-attention layers. The use of an SSM/Mamba-style backbone for cross-modal interaction is timely and potentially beneficial in terms of efficiency and stability. Empirically, the method demonstrates consistent improvements on RGB-D Scenes V2 and 7-Scenes, and the additional KITTI evaluation in the appendix strengthens the generalization claim by showing competitive performance in outdoor settings. Overall, the method is technically sound, clearly implemented, and likely useful for detection-free I2P pipelines.

Weaknesses

1.	Novelty requires clearer positioning.
Although the Focus–Sweep architecture is presented as conceptually novel, it shares similarities with existing coarse-to-fine registration paradigms. The paper does not sufficiently clarify how the Focus and Sweep components differ from prior attention-based or hierarchical matching modules, such as cross-modal fusion mechanisms based on geometric approximation. A more explicit comparison—both conceptual and empirical—would help delineate the distinct contributions of the proposed components.

2.	Limited theoretical support.
The paper provides intuitive explanations for the Focus formulation and Sweep procedure but lacks deeper theoretical analysis or structured empirical justification of key components (e.g., the scaling coefficients $\alpha, \beta, \gamma$, the hidden state update mechanism in the SSM module, and the reinforcement-style loss). Without either theoretical grounding or systematic visualization/diagnostic studies, it is difficult to assess the necessity and principled design of these elements.

3.	Experimental scope remains limited.
The main paper evaluates the method only on indoor RGB-D benchmarks. Although the appendix includes KITTI results, a more thorough evaluation in complex outdoor settings or under stronger domain shifts would strengthen claims about robustness and generalization.

---

> ### Author Rebuttal · Authors · 2026-03-30
>
> We sincerely thank the reviewer for valuable comments. Due to space limitations, many issues can only be addressed briefly here, and more detailed discussions will be included in the final version.
>
> **1.  About focus**:   The Focus module differs from transformer-based interaction by performing global feature modulation instead of attention-based token matching, using pooled point cloud statistics to generate scale and bias factors. The parameters (α, β, γ) are dynamically predicted via a learnable projection and optimized end-to-end.
>
> **2. About reinforcement-style loss**:  We clarify that the reinforcement loss refers to the reinforce-based policy loss used for dynamic interaction-depth selection, rather than any top-k selection process. In particular, hierarchical top-k selection (Section 1.3.1) is not part of the reinforcement learning module (Section 3.2). Instead, hierarchical top-k selection is used to obtain coarse-level correspondences. The key difference from previous selection methods is that prior approaches typically use a single point cloud feature to match multi-scale image features for selection, whereas our method computes three similarity maps between multi-level interacted point cloud features and multi-scale image features, and then takes the maximum value at each position across the three similarity maps to determine the coarse matches.
>
> We give more explanation of the reinforcement loss below. Given a sampled action $a$ (the selected interaction depth) from the policy $\pi_\theta(a \mid s)$ and the corresponding matching loss $L_i$, the reward is defined as:
> $$
> R = \frac{1}{L_i + \delta},
> $$
> where $\delta$ is a small constant introduced for numerical stability.
>
> The policy loss is formulated as
> $$
> L_r = - (R - B)\log \pi_\theta(a \mid s),
> $$
> where $B$ is a moving-average baseline used to reduce variance.
>
> This loss is different from standard classification or regression losses, as it optimizes a discrete interaction-depth decision through policy gradients rather than directly supervising features or labels. In this setting, the reward is derived from the inverse matching loss and reflects the overall alignment quality. If the reinforcement loss is removed, the method can be regarded as using a fixed interaction depth. Please also refer to our response to Reviewer 8sSD, Comment 2 for a related explanation.
>
> We also conducted an ablation study on the hyperparameter $\delta$, and the results are shown below:
>
> | $\delta$ | IR ↑ |
> | - | - |
> | 1e-5 | 39.9 |
> | 1e-6 | 40.1 |
> | 1e-7 | 39.8 |
>
> **3.  About outdoor**:  Due to the differences in point cloud acquisition between outdoor and indoor scenes, as illustrated in https://i.ibb.co/b4WYTfC/re1.png. The results on more outdoor datasets are shown in the https://i.ibb.co/V0MppmXS/nuscenes.png, while the generalization experiments can be found in our response to Reviewer 8sSD, Comment 4.
>
> **4.  About others**:
> - yes.
> - Given a 3D point $X$, its projection onto the image plane is given by:
> $$
> \mathbf{x} \sim \mathbf{K}(\mathbf{R}\mathbf{X} + \mathbf{t})
> $$
> where $\mathbf{K}$ is the camera intrinsic matrix and $\mathbf{x}$ is the corresponding pixel coordinate in the image.
> - We further illustrate this point with https://i.ibb.co/KjfR1JZx/re3.png. In the first example, by overcoming the scale ambiguity, the originally incorrect matches in the cup region are corrected. In the second example, after aligning the scale of the head model, the number of matches increases. In the third example, with scale information available, the matches are no longer concentrated in the front half of the staircase, but instead become correctly and more evenly distributed.
> - should be 3C.
> - We conducted additional ablation studies, where * denotes the current setting used in our method. As shown in https://i.ibb.co/HT3N8yxq/ablation.png.
> - We will provide further theoretical analysis in the follow-up and final version, interpreting Focus as conditional feature modulation, Sweep as structured sequence modeling, and RL-based depth selection as discrete optimization.
> - We clarify that $F_I \in \mathbb{R}^{N_I \times C}$ and $F_{P} \in \mathbb{R}^{N_P \times C}$ share the same feature dimension $C$. Here, $N_I$ and $N_P$ denote the total numbers of image and point cloud features after concatenating the three hierarchical levels. Cosine similarity result in $S_k \in \mathbb{R}^{N_I \times N_P}$.
> - The reinforcement learning loss is defined in Eq. (11).
> - We added yellow boxes to highlight the key regions, as shown in https://i.ibb.co/4g3cRgyc/re4.png. It can be seen that, in the main scene structures and objects, the dense variation regions of the point cloud are largely consistent with the image contours, which indicates that our method estimates the pose accurately.
> - We conducted, where HTS denotes hierarchical  top-k selection process.
> - HTS stands for hierarchical top-k selection, and DLAS stands for dynamic layer allocation strategy.

---

> > ### Author Rebuttal · Reviewer_dZFR · 2026-04-01
> >
> > I am willing to raise my score with the newly added experiments and figures, as long as the revision intensity is within the conference allowance.

---

> > > ### Author Response · Authors · 2026-04-01
> > >
> > > We sincerely appreciate the time, effort, and constructive comments you have dedicated to reviewing our manuscript.
> > > We are happy to address any further questions you may have and will include our previous responses in the final version.
> > > In light of the thorough revisions we have made based on your insightful feedback, we are confident that the paper has been significantly improved.
> > >
> > > We therefore sincerely hope that you will kindly reconsider and favorably adjust your score for this manuscript.
> > >
> > > Thank you again for your thoughtful review and valuable guidance.
> > >
> > >  ：）

---

### Official Review · Reviewer_8sSD · 2026-03-09

**Soundness:** 2
**Presentation:** 2
**Significance:** 2
**Originality:** 3
**Overall Recommendation:** 4
**Confidence:** 3

**Summary:**

The paper studies image-to-point-cloud registration, which aims to estimate the relative pose of an image given a 3D point cloud. The authors propose FS-I2P, a Mamba-based network for learning cross-modal correspondences. The model includes a hierarchical Focus–Sweep interaction mechanism, where the Focus module performs global conditioning of image features using point-cloud information, and the Sweep module performs cross-modal interaction through a Hierarchical Token Sequence (HTS). The paper also introduces a Dynamic Layer Allocation Strategy that uses reinforcement learning to adaptively determine the number of interaction layers during inference. Experiments on RGB-D Scenes and 7-Scenes show improvements over prior methods.

**Compliance With Llm Reviewing Policy:**

Affirmed.

**Final Justification:**

I have read the author's response and the other reviewers' comments. I appreciate the author's efforts, and I believe the rebuttal has adequately addressed my previous concerns through additional comparisons and more detailed explanations of the implementation. I am willing to raise my score.

**Key Questions For Authors:**

1. Controlled comparison with transformer architecture. Could the authors provide a controlled comparison between the proposed SSM-based Sweep module and a transformer-based cross-attention module, keeping tokenization, depth, and parameter counts comparable? This would help clarify whether the observed improvements are due to the SSM mechanism itself or other architectural choices.
2. Can the authors provide further analysis of the dynamic layer allocation policy, such as the distribution of the learned interaction depths and performance comparisons with fixed-depth variants?
3. Sequential token design. The SSM interaction relies on sequential, order-dependent token processing, which is not typical in correspondence problems. Is making the tokens sequential necessary or beneficial, or is it primarily a constraint imposed by the Mamba architecture? How sensitive is the method to different token ordering strategies in the Hierarchical Token Sequence?

**Limitations:**

It would be helpful for the authors to discuss aspects such as the generalization ability of the method across scenes with varying complexity and scale of scenes, as well as computational constraints in large-scale environments. In addition, simplifying the narrative could make it easier for readers to grasp the core technical contributions more quickly.

**Strengths And Weaknesses:**

**Strength**
- The paper addresses the problem of image-to-point-cloud registration, which is important for applications such as camera relocalization, robotics, augmented reality
- The proposed Focus–Sweep architecture follows a coarse-to-fine strategy: global conditioning of image features followed by localized cross-modal interaction. This design is intuitive and aligns with common design patterns in multimodal correspondence estimation.
- The usage of SSM-based interaction from Mamba may offer computational benefits due to its linear complexity.

**Weakness**
- Soundness. The claim that SSM-based scanning provides advantages over transformer attention is not fully convincing (beyond potential efficiency benefits). The comparison with transformer-based models is limited and not sufficiently controlled. Similar interaction patterns could likely be implemented using transformer variants, such as window-based attention (e.g., Swin Transformer) or cross-attention mechanisms.
- Missing related work. The problem setting of I2P is closely related to camera relocalization, but the paper does not sufficiently discuss relevant literature such as scene coordinate regression methods (e.g., DSAC [1], SCRNet [2]). These methods also estimate camera pose given a 3D scene, but they are neither discussed nor compared in the paper.
- Presentation. The human-inspired cognition motivation (Mamba, Focus and Sweep) is somewhat overstated and distracting. Similar ideas such as global feature conditioning and cross-modal interaction have been explored in prior 2D–2D and 2D–3D correspondence works, and the novelty of this is not fully clear.



[1] Brachmann, Eric, et al. "Dsac-differentiable ransac for camera localization." Proceedings of the IEEE conference on computer vision and pattern recognition. 2017.
[2] Li, Xiaotian, et al. "Hierarchical scene coordinate classification and regression for visual localization." Proceedings of the IEEE/CVF Conference on Computer Vision and Pattern Recognition. 2020.

---

> ### Author Rebuttal · Authors · 2026-03-30
>
> We appreciate this valuable suggestion. In the revision, we will include a brief discussion of camera relocalization methods (e.g., DSAC and SCRNet) to better position our work within the related literature and clarify the differences between these approaches and our I2P setting.
> We will also refine the presentation of our human-inspired motivation to make it more concise and less distracting, highlighting the specific novelty of our approach. We will address the main concerns raised below.
>
> **1.  About comparison with transformer**:  We designed a fully consistent sweep-focus framework, where the Mamba layers were replaced with a standard Transformer and a Swin Transformer. All experiments were conducted on RGB-D V2 with a fixed depth of two layers.
>
> |Method|RR|memory|time|
> |-|-|-|-|
> |mamba|69.9|6327|0.312|
> |transformer|65.8|8134|0.502|
> |swim-transformer|69.7|7932|0.481|
>
> These results provide several important insights. First, compared with the standard Transformer, Mamba achieves clear gains in both efficiency and performance, reducing memory usage and inference time while also improving RR. This indicates that the improvements are closely related to the SSM-based sequence modeling mechanism, which enables efficient long-range dependency modeling without quadratic complexity.
>
> Second, compared with Swin-Transformer, which introduces window-based local interactions, Mamba achieves slightly better performance (69.9 vs. 69.7) while maintaining significantly lower computational cost. Notably, the sweep–focus framework itself is naturally compatible with window-style interaction patterns, as the sequential scanning and aggregation process aligns well with localized token grouping. This explains why Swin-Transformer performs competitively in this framework. However, Mamba still provides both higher accuracy and better efficiency, demonstrating a more favorable overall trade-off.
>
> Overall, under controlled settings, these results demonstrate that the proposed SSM-based Sweep module offers a more efficient alternative to cross-attention-based designs, while maintaining strong performance. This supports our claim that the improvements stem from the intrinsic properties of the SSM mechanism, rather than other architectural factors.
>
> **2.  About further analysis of the dynamic layer allocation policy**:  We have added comparisons at a fixed scale and examined the choice of layer numbers under different mean depths.
> For clarity, we use scale to denote the number of FS-layers.
>
> |Scale|RR↑|
> |-|-|
> |0|56.4|
> |1|63.2|
> |2|68.2|
> |3|67.7|
> |Dynamic(Ours)|72.9|
>
> The results indicate that, among the static configurations, a fixed interaction depth of 2 achieves the best performance. Performance improves as the depth increases from 1 to 2, but degrades at depth 3, suggesting that excessive interaction may introduce noise and hinder learning.
> In contrast, the proposed dynamic strategy further enhances performance, achieving the best result of 72.9 and outperforming all fixed-depth variants. This demonstrates that different samples benefit from different interaction depths, highlighting the advantage of adaptively adjusting the interaction depth for each sample.
>
>
> |Percentage/%|Scale=0|Scale=1|Scale=2|Scale=3|
> |-|-|-|-|-|
> |Depth1(1.74)|16.5|24.33|31.83|27.33|
> |Depth2(1.66)|19.5|22.33|30.33|27.83|
> |Depth3(1.18)|17.83|22.67|34.5|25.0|
> |Depth4(1.39)|21.83|21.17|31.83|25.17|
>
> The learned depth distributions are well balanced, with all depth levels being utilized. In particular, depth=2 is selected most frequently, while depth=0 also maintains a non-negligible proportion, indicating that the policy learns to skip unnecessary interactions for simpler cases. This demonstrates that the dynamic allocation does not collapse to a single depth, but instead adapts to sample difficulty.
>
> **3.  About  sequential token design**:  Our design mainly exploits the interactive properties of Mamba to refine the interaction process, and we also evaluated different ordering strategies. RasterOrder arranges tokens in a row-wise manner from left to right and top to bottom. ReverseOrder arranges tokens in the opposite direction, from bottom to top and right to left. Column-wise arranges tokens column by column, from top to bottom and left to right. We use IR as the evaluation metric because it directly reflects the quality of cross-modal matching and retrieval, which is most relevant to the effectiveness of the sequential token design.
>
>
> |Strategy|IR↑|
> |-|-|
> |RasterOrder|38.8|
> |ReverseOrder|35.8|
> |Column-wise|37.5|
> |Ours|40.1|
>
> **4.  About  generalization ability of the method across scenes with varying complexity and scale of scenes**:
> We also conducted additional experiments on more outdoor scenes and generalization.
>
> |Model%|IR↑|FMR↑|RR↑|
> |-|-:|-:|-:|
> |7-Scenes→RGB-DScenesv2||||
> |2D3D-MATR|17.5|66.2|19.8|
> |FS-I2P|19.2|68.6|24.4|
>
>
> Due to space limitations, more outdoor experimental results are provided at: https://i.ibb.co/V0MppmXS/nuscenes.png

---

> > ### Author Rebuttal · Reviewer_8sSD · 2026-04-03
> >
> > I have read the author's response and the other reviewers' comments. I believe the rebuttal addresses my concerns, and I am willing to raise my score.

---

> > > ### Author Response · Authors · 2026-04-03
> > >
> > > We sincerely appreciate the time, effort, and constructive feedback you have devoted to reviewing our manuscript. We will incorporat our detailed responses into the revised version.
> > >
> > > Thank you again for your recognition of our work and your valuable feedback, it means a lot to us.
> > >
> > > :)

---

### Decision · Program_Chairs · 2026-04-30

**Decision:**

Accept (regular)

**Comment:**

This submission eventually got four positive recommendations. Initially, the reviewers were concerned about the novelty, the presentation, and the soundness of the technical solutions. The authors did a good job and addressed most of these concerns in the rebuttal. During the discussion among the authors and the reviewers, the reviewers confirmed that their concerns had been fully addressed. Thus, all reviewers reached a consensus without a discussion. The AC read through the manuscript, all reviews, the rebuttal, the discussions among the authors and the reviewers, and the author AC confidential comments, the AC agreed with all reviewers, and liked the idea of the paper. Per these, the AC made a decision of acceptance. This decision was approved by the SAC as well.